# Identification of a Novel Two-Peptide Lantibiotic from *Vagococcus fluvialis*

Zuzana Rosenbergová,[a,b] Thomas F. Oftedal,[a] Kirill V. Ovchinnikov,[a] Thasanth Thiyagarajah,[a] Martin Rebroš,[b] Dzung B. Diep[a]

[a]Faculty of Chemistry, Biotechnology and Food Science, Norwegian University of Life Sciences, Ås, Norway
[b]Institute of Biotechnology, Faculty of Chemical and Food Technology, Slovak University of Technology, Bratislava, Slovakia

Zuzana Rosenbergová and Thomas F. Oftedal contributed equally to this work as first authors. Author order was determined by mutual agreement.

**ABSTRACT** Infections caused by multiresistant pathogens have become a major problem in both human and veterinary medicine. Due to the declining efficacy of many antibiotics, new antimicrobials are needed. Promising alternatives or additions to antibiotics are bacteriocins, antimicrobial peptides of bacterial origin with activity against many pathogens, including antibiotic-resistant strains. From a sample of fermented maize, we isolated a *Vagococcus fluvialis* strain producing a bacteriocin with antimicrobial activity against multiresistant *Enterococcus faecium*. Whole-genome sequencing revealed the genes for a novel two-peptide lantibiotic. The production of the lantibiotic by the isolate was confirmed by matrix-assisted laser desorption ionization–time of flight (MALDI-TOF) mass spectrometry, which revealed distinct peaks at 4,009.4 *m/z* and 3,181.7 *m/z* in separate fractions from reversed-phase chromatography. The combination of the two peptides resulted in a 1,200-fold increase in potency, confirming the two-peptide nature of the bacteriocin, named vagococcin T. The bacteriocin was demonstrated to kill sensitive cells by the formation of pores in the cell membrane, and its inhibition spectrum covers most Gram-positive bacteria, including multiresistant pathogens. To our knowledge, this is the first bacteriocin characterized from *Vagococcus*.

**IMPORTANCE** Enterococci are common commensals in the intestines of humans and animals, but in recent years, they have been identified as one of the major causes of hospital-acquired infections due to their ability to quickly acquire virulence and antibiotic resistance determinants. Many hospital isolates are multiresistant, thereby making current therapeutic options critically limited. Novel antimicrobials or alternative therapeutic approaches are needed to overcome this global problem. Bacteriocins, natural ribosomally synthesized peptides produced by bacteria to eliminate other bacterial species living in a competitive environment, provide such an alternative. In this work, we purified and characterized a novel two-peptide lantibiotic produced by *Vagococcus fluvialis* LMGT 4216 isolated from fermented maize. The novel lantibiotic showed a broad spectrum of inhibition of Gram-positive strains, including vancomycin-resistant *Enterococcus faecium*, demonstrating its therapeutic potential.

**KEYWORDS** bacteriocin, *Vagococcus*, lantibiotic, antimicrobial, vagococcin T, pore formation

Address correspondence to Dzung B. Diep, dzung.diep@nmbu.no.

The authors declare no conflict of interest.

Enterococci such as *Enterococcus faecium* and *E. faecalis* are regular commensals of human and animal intestines (1, 2). However, in recent years, enterococci have become a concern in both human and veterinary medicine as they have emerged as some of the most prevalent nosocomial pathogens (3, 4). In addition to their ability to effectively acquire, harbor, and distribute antimicrobial resistance (AMR) determinants, enterococci are robust and able to survive on nonbiotic surfaces for prolonged periods

(5, 6). There is increasing evidence that the overuse of antibiotics is a primary selection pressure for the acquisition and dissemination of antibiotic resistance in bacteria (7). To reduce the dissemination of AMR and to combat resistant bacteria, alternatives to antibiotics are needed. One such promising alternative is bacteriocins, natural protein-aceous compounds produced by bacteria with antimicrobial activity mostly against closely related species, including pathogenic and antibiotic-resistant strains.

Small bacteriocins (<10 kDa) are classified based on their biosynthesis: posttranslation-ally modified bacteriocins belong to class I, while unmodified bacteriocins are members of class II (8, 9). Lanthipeptides, which belong to class I, are characterized by thioether link-ages formed between cysteines and dehydrated serine and threonine residues to yield lanthionine and methyllanthionine, respectively (10). The organization of the ring struc-tures then recognizes a specific target on sensitive cells, such as lipid II, which is the dock-ing molecule for most lantibiotics (11). The bacteriocin producer must protect itself from the lethal action of its own bacteriocin. For lantibiotics, self-immunity is achieved by the production of immunity proteins commonly named LanI and/or LanFE(G) (12, 13). The LanFE(G) proteins compose a specialized ABC transporter that mediates the efflux of mature lanthipeptides from the cell, while LanI is thought to protect the producer extrac-ellularly against the secreted lanthipeptide (12).

Lantibiotics are further subdivided into at least two types based on differences in the modification enzymes (14). Type I lanthipeptides, of which nisin is the founding member, use two separate enzymes for the dehydration (LanB) and cyclization (LanC) steps that produce the (methyl)lanthionine rings. Type II employs a single bifunctional enzyme (LanM) that catalyzes both steps (10, 14). LanM modification enzymes usually carry out the modification of two-peptide lantibiotics, each of which consists of two different pep-tides exhibiting considerable synergy when combined but having little or no activity when assessed individually (15). The most well-studied two-peptide lantibiotic, lacticin 3147, has potent activity against numerous pathogenic Gram-positive species, including vancomycin-resistant enterococci (VRE) (16). Lacticin 3147 also attenuates the growth of *Staphylococcus aureus* in a murine infection model and disrupts *Streptococcus mutans* biofilms, demonstrating the clinical potential of lantibiotics (17, 18).

*Vagococcus fluvialis* belongs to a genus of motile lactic acid bacteria most closely related to *Enterococcus* and *Carnobacterium* and was first described as a phylogenetically distinct genus in 1989 (19, 20). Not much is known about *V. fluvialis*; most characterized isolates originated from wounds of animals (pigs, horses, and cattle) and from human clinical cases (20, 21). However, the species has also been isolated from the urine of healthy cattle and was described as a potential probiotic in fish (22, 23). In this work, we describe the discovery and characterization of a novel two-peptide lantibiotic produced by *Vagococcus fluvialis* LMGT 4216. The bacteriocin was active against most Gram-posi-tive strains tested, including animal and human pathogens, such as multidrug-resistant *E. faecium* and mastitis-associated *Streptococcus uberis* (24). The bacteriocin gene cluster had an atypical organization and included what resembles a quorum-sensing system. To our knowledge, this is the first bacteriocin characterized from *Vagococcus*. We believe that this bacteriocin could serve an important role as a therapeutic in the future.

## RESULTS

**Screening for bacteriocin producers against *Enterococcus faecium*.** *E. faecium* LMG 20705 is a multidrug-resistant opportunistic pathogen. The resistance pattern was determined by AMRFinderPlus (see Table S1 in the supplemental material) and a disc diffusion test according to EUCAST methods (data not shown) (25, 26). The strain was shown to be resistant to vancomycin, ampicillin, and streptomycin, all of which are first-line therapeutics for enterococcal infections (27). In addition, the strain exhibited resistance to quinupristin-dalfopristin, a mixture of streptogramins B and A used for the treatment of serious VRE-related infections (28).

A total of 40 different samples of fermented fruits and vegetables were screened for the presence of bacteriocin producers that could inhibit the growth of *E. faecium*

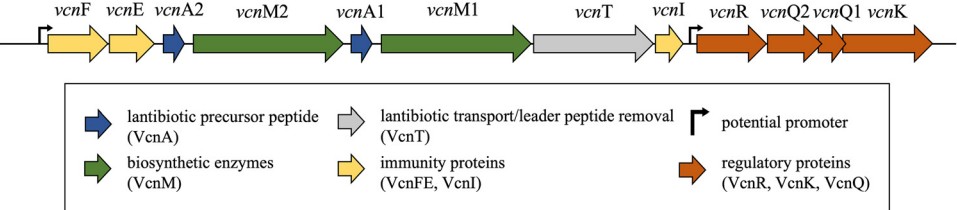

**FIG 1** Gene organization of the vagococcin T cluster in *V. fluvialis* LMGT 4216. Bifunctional modification enzyme genes (green) are located downstream of lantibiotic precursor genes (blue). A lantibiotic transporter gene with a leader removal function (gray) is located downstream of *vcn*M1 and upstream of *vcn*I, encoding a potential immunity protein (yellow). Other genes involved in bacteriocin immunity are located at the beginning of the cluster. A group of genes resembling a quorum-sensing system (red) is located at the end of the cluster.

LMG 20705. From all samples, 17 colonies exhibited a distinct inhibition zone indicative of antimicrobial production. Repetitive element PCR (rep-PCR) was performed to examine the genetic similarity of these isolates. Nine unique DNA band profiles were observed after gel electrophoresis (data not shown). One representative from each group was selected for whole-genome sequencing to identify novel bacteriocin genes. The genomes were analyzed for bacteriocins by BAGEL4 and antiSMASH (29, 30). The analysis revealed that all but two isolates had genes for previously characterized bacteriocins known to be active against enterococci: subtilosin A (31), ericin S (32), enterolysin A (33), and NKR-5-3B (34). One of the two isolates with a potentially novel bacteriocin was a strain isolated from fermented maize; the genome of this isolate contained a gene cluster with an organization similar to those of the two-peptide lantibiotic gene clusters. The best database hit for the predicted bacteriocin was the lantibiotic flavecin from *Ruminococcus flavefaciens* (35), with only 45% identity, suggesting that the isolate, identified as *Vagococcus fluvialis*, likely produced a novel two-peptide lantibiotic.

**Genome analysis and identification of the vagococcin T gene cluster.** The search for putative bacteriocin genes by antiSMASH resulted in the identification of a type II lantibiotic gene cluster (Fig. 1). Two bacteriocin genes, *vcn*A1 and *vcn*A2, were identified and predicted to represent the $\alpha$ (*vcn*A1) and $\beta$ (*vcn*A2) peptides of a two-peptide lantibiotic hereafter named vagococcin T (Vcn T$\alpha$ and Vcn T$\beta$). Located downstream of each of the *vcn*A1 and *vcn*A2 genes are genes encoding lantibiotic biosynthesis proteins, *vcn*M1 and *vcn*M2, respectively. Both gene products, VcnM1 and VcnM2, showed sequence similarity with MrsM, the modification enzyme for the lantibiotic mersacidin (36). The predicted functions of all proteins encoded by the *vcn* gene cluster are listed in Table 1.

**TABLE 1** Encoded proteins from the vagococcin T cluster of *V. fluvialis* LMGT 4216 with their homologs and predicted functions

| Gene product | Putative function(s) | Homolog, % sequence identity (GenBank accession no.) |
|---|---|---|
| VcnF | Bacteriocin immunity | NisF, 47 (AAC43327.1) |
| VcnE | Bacteriocin immunity | MrsE, 22 (CAB60257.1) |
| VcnA2 | Vagococcin T $\beta$-peptide | FlvA2b, 46 (P0DQL4.1) |
| VcnM2 | VcnA2 dehydratase and cyclase | MrsM, 44 (CAB60261.1) |
| VcnA1 | Vagococcin T $\alpha$-peptide | FlvA1a, 42 (P0DQM1.1) |
| VcnM1 | VcnA1 dehydratase and cyclase | MrsM, 48 (CAB60261.1) |
| VcnT | Bacteriocin maturation and export | MrsT, 45 (KAF1340276.1) |
| VcnI | Bacteriocin immunity | |
| VcnR | Response regulator | FsrA, 39 (EIA6660097.1) |
| VcnQ2 | Pheromone maturation/export | FsrB, 35 (EGO8521395.1) |
| VcnQ1 | Pheromone/signaling molecule prepeptide | FsrD, 37 (CDK37795.1) |
| VcnK | Protein histidine kinase | FsrC, 35 (EIP8082021.1) |

**A**

**VcnA1**

MERNPILREKKQQQFLSTSGLEEVNQNIEFIENLSGG   NNVWVTILQGVVGCVASWAVGNKGKVCTWTVECQKNCS

**VcnA2**

MRTSNDIKNKTGYVEESKLKEMIEEPDYSGG   AWTTLPCIGGIIAATLNFDACPTSACTKSCNK

**B**

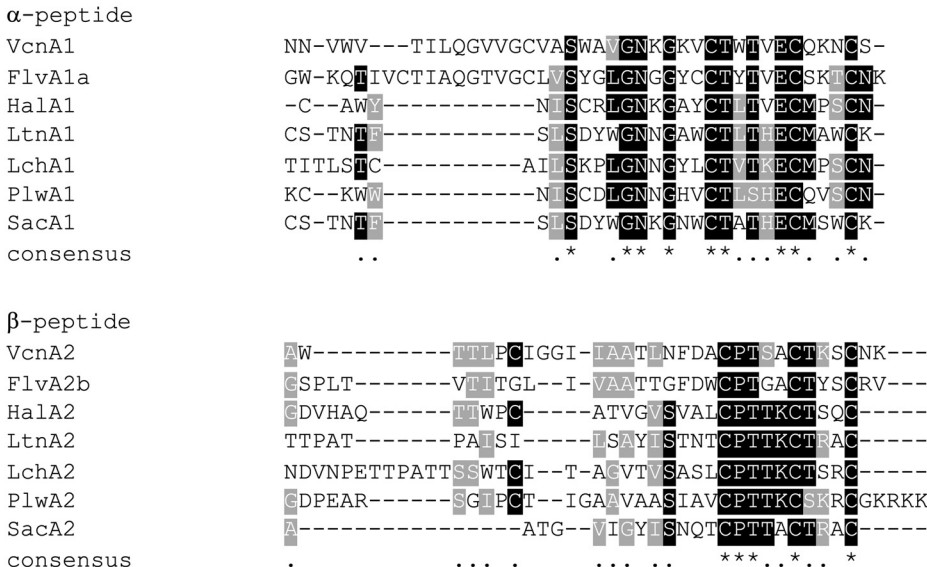

FIG 2 (A) Predicted amino acid sequences of vagococcin T prepeptides. Leader sequences are underlined and separated from the mature peptides by a space. (B) Multiple-sequence alignment of α- and β-peptides of the known two-peptide lantibiotics flavecin (Flv) (UniProt accession numbers P0DQM1 and P0DGL4), haloduracin (Hal), lacticin 3147 (Ltn) (accession numbers O87236 and O87237), lichenicidin (Lch) (accession numbers P86475 and P86476), plantaricin W (Plw) (accession numbers D2KR94 and Q9AF68), and staphylococcin C55 (Sac) (accession numbers Q9S4D3 and Q9S4D2). The sequence alignment was performed using T-Coffee and colored with BoxShade; black and gray shading correspond to identical and similar amino acids, respectively.

The vcnT gene is located downstream of vcnM1 and encodes a C39 peptidase that shows 45% identity with MrsT, the mersacidin transport enzyme that cleaves the leader after the GG/GA motif, a typical cleavage site for many bacteriocin leaders (37). A GG motif is indeed present in both the VcnA1 and VcnA2 prepeptides (Fig. 2A). The mature peptides showed the highest homology to the flavecin FlvA1a and FlvA2b peptides (42% and 46%, respectively) (35). Sequence alignment of Vcn Tα with other lantibiotic α-peptides (Fig. 2B) showed that Vcn Tα contains the same conserved CTxTxEC motif believed to be essential for lipid II docking (38). Similarly, the conserved sequence (CPTxxCt/sxxC; variable residues are shown in lowercase, threonine/serine) typical for all β-peptides was found in Vcn Tβ (Fig. 2B).

The types of immunity genes present in lantibiotic gene clusters vary, and the encoded immunity proteins often show little sequence identity with each other (39). Two genes of the LanFE(G) immunity system are present in the vcn cluster, vcnF and vcnE, located at the start of the operon. VcnF showed 47% identity to the ATP-binding domain NisF of the NisFEG transporter and contained the conserved sequences for both Walker A and B motifs (40).

The last four genes in the cluster resembled an analog of the Fsr quorum-sensing system of E. faecalis; this type of quorum-sensing system has not previously been identified in other lantibiotic clusters (41). The product of the first open reading frame (ORF), designated vcnR, showed 39% identity to the response regulator (RR) FsrA (Table 1). An FsrB homolog is encoded by the gene designated vcnQ2, with 36% identity (Q for quorum). The third component, a sensor histidine protein kinase (HPK)

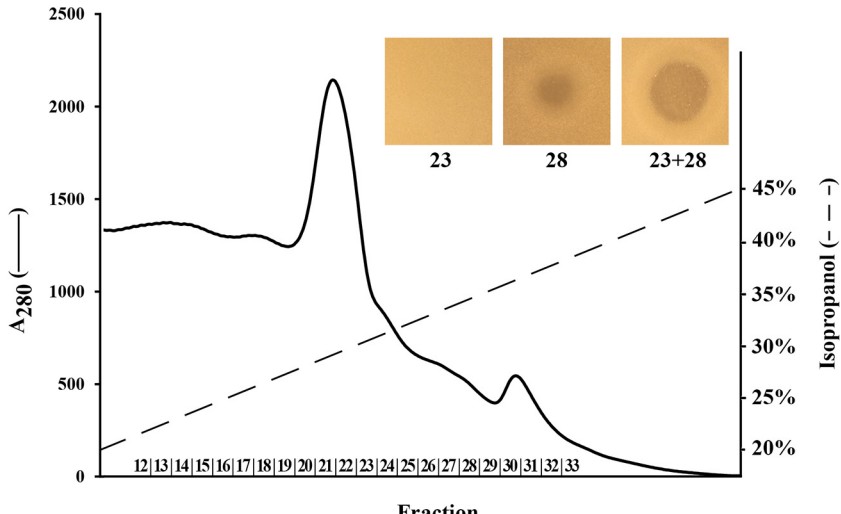

**FIG 3** Reversed-phase chromatography elution profile of the sample obtained by cation-exchange chromatography. All collected fractions exhibited relatively low bacteriocin activity against *E. faecium* LMG 20705, but two fractions showed a significant increase in potency when assayed together (1:1 [vol/vol]), indicating the presence of a two-peptide bacteriocin. The inhibition of *E. faecium* LMG 20705 by individual and combined fractions is pictured in the top right corner.

encoded by *vcn*K, showed 35% identity to FsrC. A search for small reading frames that could encode the pheromone component of the quorum-sensing system revealed a small ORF between *vcn*Q2 and *vcn*K. The product of this ORF gave no hits to any known peptides by a BLAST search; however, sequence alignment showed 37% identity to FsrD, the gelatinase biosynthesis-activating pheromone (GBAP) prepeptide (42). It is therefore possible that the processed product of *vcn*Q1 is a pheromone.

Another small ORF located between *vcn*T and *vcn*R also showed no sequence homology to known proteins by a BLAST search; however, the gene product had a size, charge, and hydrophobicity similar to those of known lantibiotic immunity proteins. LanI proteins with comparable physicochemical properties include EciI, PepI, and LasJ, the LanI immunity proteins for epicidin 280, Pep5, and lactocin S (39). The ORF located between *vcn*T and *vcn*R was therefore named *vcn*I and is further discussed in Discussion below.

Because of the novelty of vagococcin T, the antimicrobial produced by the isolate of *V. fluvialis*, named *V. fluvialis* LMGT 4216 here, was chosen for further characterization. Cell-free supernatants from the isolate contained an antimicrobial substance that was heat stable and sensitive to proteinase K (data not shown), properties expected for bacteriocins like vagococcin T (8).

**Purification of bacteriocin.** The purification of the predicted two-peptide lantibiotic produced by *V. fluvialis* LMGT 4216 was achieved by a three-step purification scheme consisting of ammonium sulfate precipitation, cation-exchange chromatography, and reverse-phase chromatography (RPC) (43). During the RPC elution, two peaks corresponding to 29% and 36% isopropanol were observed in the elution profile (Fig. 3). The collected fractions were assayed against *E. faecium* LMG 20705; a low antimicrobial activity of 400 bacteriocin units (BU)/mL was found only in the second peak (fractions 26 to 30), which would be expected due to the separation of the two peptides into separate fractions (44). To test this notion, fractions 21 to 24 were individually combined with fractions 26 to 30 in a 1:1 (vol/vol) ratio to find any combination of fractions exhibiting synergy (Fig. S1). Indeed, the highest synergy was observed between fractions 23 and 28, which in combination had an antimicrobial activity of 51,200 BU/mL, representing a 1,200-fold increase in activity with a yield of 128% (Table 2).

With purified bacteriocin, the biological activity of vagococcin T against a number of bacteria was determined using a spot-on-lawn assay (Table 3). Lantibiotics are known to

**TABLE 2** Bacteriocin purification

| Sample | Vol (mL) | Activity (BU/mL) | Total activity (BU) | Yield (%) |
|---|---|---|---|---|
| Supernatant | 1,000 | 80 | 80,000 | 100 |
| Ammonium sulfate precipitate | 150 | 320 | 48,000 | 60 |
| Cation-exchange chromatography | 100 | 160 | 16,000 | 20 |
| Reversed-phase chromatography | 2 | 51,200 | 102,400 | 128 |

be very potent against Gram-positive bacteria but have limited activity against Gram-negative bacteria, as observed for nisin (45), lichenicidin (46), and thusin (47). In addition to showing potent antimicrobial activity against the indicator strain *E. faecium* LMG 20705, vagococcin T displayed a broad inhibition spectrum, including all Gram-positive bacteria tested except for *Staphylococcus aureus*. The Gram-negative bacteria *Escherichia coli* and *Salmonella enterica* serovar Typhimurium were not inhibited, which is expected for the lipid II-targeting type A lantibiotics.

**Molecular mass and bacteriocin identification.** Given the synergism of fraction 23 with fraction 28, these fractions were analyzed further using matrix-assisted laser desorption ionization–time of flight mass spectrometry (MALDI–TOF MS). The acquired spectra revealed the presence of only one distinct peak in each fraction. A peak at 4,009.5 *m/z* can be seen in fraction 23 (Fig. 4A), which correlated well with the mass predicted for one of the two peptides by antiSMASH (30) (assuming 1 unmodified Ser/Thr residue). The peak in fraction 28 (Fig. 4B) at 3,181.7 *m/z*, however, differed from the prediction by 70.1 Da (3,111.6 Da, assuming 2 unmodified Ser/Thr residues). The reasoning behind this difference is given in Discussion below. A schematic representation of all posttranslational modifications of Vcn Tα and Vcn Tβ consistent with the measured masses is shown in Fig. 5. These results confirm that the antimicrobial activity produced by *V. fluvialis* LMGT 4216 was indeed caused by the predicted two-peptide lantibiotic vagococcin T.

**Pore-forming nature of vagococcin T.** To assess whether vagococcin T is a pore former, a propidium iodide (PI) assay was conducted. PI is a membrane-impermeant dye that increases its fluorescence efficiency/quantum yield when bound to double-stranded

**TABLE 3** Inhibition spectrum of reversed-phase chromatography-purified vagococcin T (2 μL)

| Indicator strain[a] | Sensitivity[b] |
|---|---|
| *Bacillus cereus* LMGT 2805 | ++ |
| *Bacillus cereus* LMGT 2731 | + |
| *Enterococcus faecalis* LMGT 2333 | ++ |
| *Enterococcus faecalis* LMGT 3331 | ++ |
| *Enterococcus faecium* LMGT 2772 | ++ |
| *Enterococcus faecium* LMGT 3104 | ++ |
| *Lactobacillus curvatus* LMGT 2353 | +++ |
| *Lactobacillus plantarum* LMGT 2352 | ++ |
| *Lactococcus garvieae* LMGT 3390 | ++ |
| *Lactococcus lactis* LMGT 2081 | ++ |
| *Listeria innocua* LMGT 2710 | +++ |
| *Listeria monocytogenes* LMGT 2604 | ++ |
| *Listeria monocytogenes* LMGT 2650 | + |
| *Pediococcus acidilactici* LMGT 2002 | +++ |
| *Streptococcus dysgalactiae* LMGT 3890 | + |
| *Streptococcus thermophilus* LMGT 3555 | +++ |
| *Streptococcus uberis* LMGT 3912 | ++ |
| *Staphylococcus haemolyticus* LMGT 4133 | + |
| *Staphylococcus aureus* LMGT 3242 | − |
| *Salmonella* Typhimurium B1377 | − |
| *Escherichia coli* TG1 | − |

[a]Laboratory of Microbial Gene Technology (LMGT), Norwegian University of Life Sciences, Ås, Norway.
[b]Inhibition zone diameters of 5 to 9 mm (+), 10 to 14 mm (++), or >15 mm (+++) or no inhibition (−).

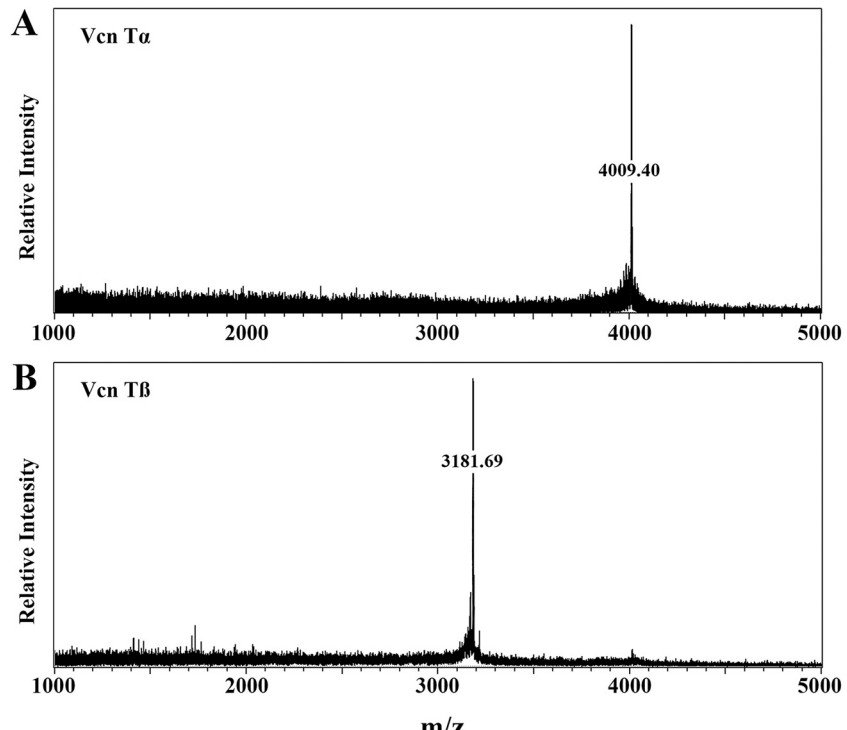

**FIG 4** MALDI-TOF mass spectrometry analysis of fractions 23 (A) and 28 (B) from reversed-phase chromatography. The peaks at 4,009.40 *m/z* and 3,181.69 *m/z* represent Vcn Tα and Vcn Tβ peptides, respectively.

DNA (48). After exposing the indicator strain to the known pore-forming lantibiotics nisin A and nisin Z in the presence of extracellular PI, an increase in the emission was detected (Fig. 6). Similar results were also obtained for vagococcin T, implying that vagococcin T has a similar mode of action involving pore formation. The negative control, micrococcin P1, a bacteriocin that kills cells by inhibiting protein synthesis (49), caused little or no increase in fluorescence as it does not form pores.

To further corroborate our results showing that vagococcin T is membrane active, the indicator cells exposed to vagococcin T were examined by scanning electron microscopy (SEM). Clear differences were observed for bacteriocin-treated compared to untreated cells (Fig. 7). Treated cells appeared collapsed/shriveled, suggesting a loss of turgor pressure. Irregular dark spots were visible on some cells, possibly indicating pores or damage to the cell envelope. In addition, an extracellular matrix-like material was visible only in the treated cells. In comparison, the cell surface of untreated cells was smooth, without ruptures or signs of cell damage.

**Stress response involved in resistance to vagococcin T.** Resistant colonies of *E. faecium* LMG 20705 were occasionally visible within the inhibition zones of vagococcin T. The increased tolerance to vagococcin T of four randomly selected spontaneous mutants was tested and showed a 64- to 256-fold increase in the MIC compared to that for the wild type (Table 4). The frequency of resistant mutants was estimated to be $8.7 \times 10^{-7}$ based on plating techniques. Whole-genome sequencing was performed on the four mutants to identify the possible mechanism for the increased tolerance to vagococcin T. Three of the four mutants had mutations in *liaF* (M1 to M3), two with nonconservative missense mutations (Ile108Asn and Trp141Ser) and one with a frameshift from amino acid position 9 (Val9fs) (M2). Several mutations were found in various genes of mutant M4, none of which could be directly linked to the increased tolerance to vagococcin T (Table 4). *liaF* encodes a negative regulator of LiaRS, a two-component regulatory system involved in the cell envelope stress response induced by lipid II-interacting antimicrobials

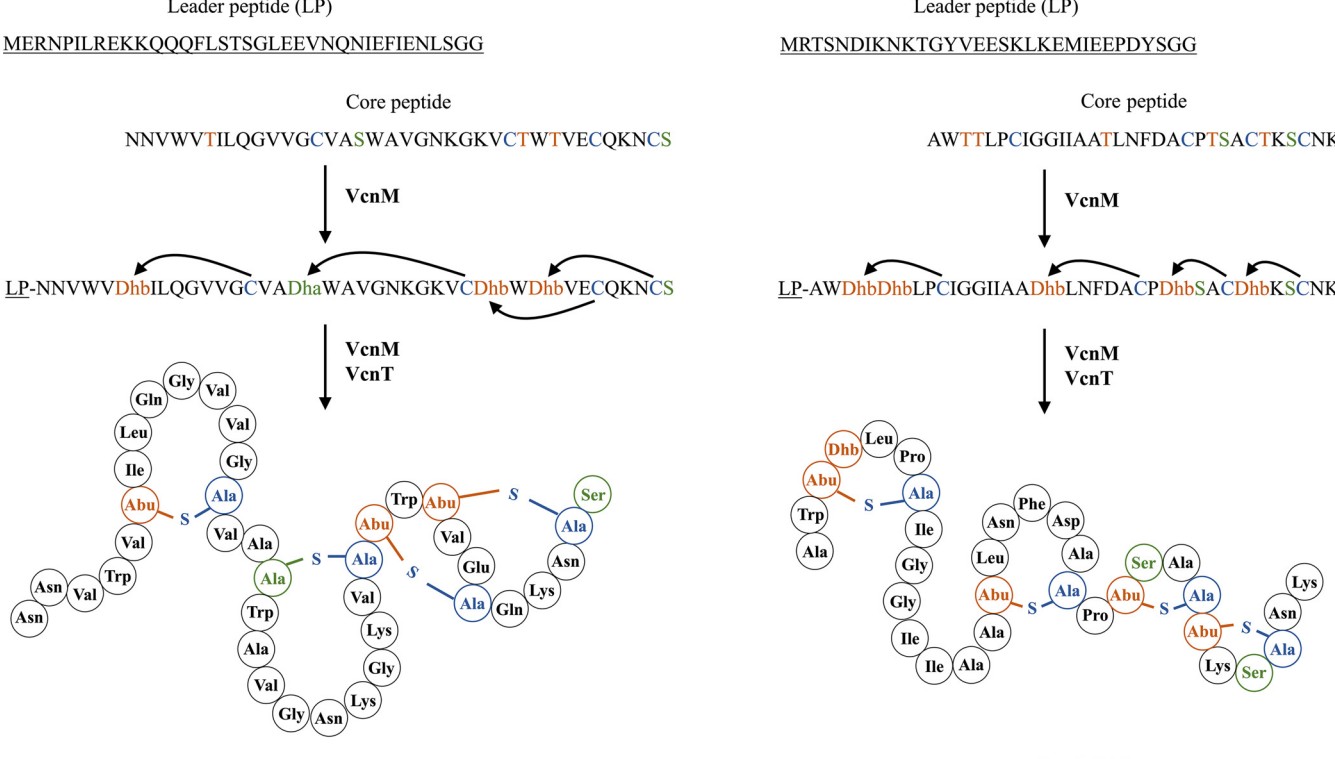

Leader peptide (LP)

MERNPILREKKQQQFLSTSGLEEVNQNIEFIENLSGG

Core peptide

NNVWVTILQGVVGCVASWAVGNKGKVCTWTVECQKNCS

**Vcn Tα**    4009.4 Da

Leader peptide (LP)

MRTSNDIKNKTGYVEESKLKEMIEEPDYSGG

Core peptide

AWTTLPCIGGIIAATLNFDACPTSACTKSCNK

**Vcn Tβ**    3181.9 Da

**FIG 5** Proposed biosynthetic scheme for vagococcin T α- and β-peptides. The structures of Vcn Tα and Vcn Tβ were deduced from the known structures of other two-peptide lantibiotics. Lanthionine rings (Ala-S-Ala) are formed between didehydroalanine (Dha), derived from serine (green) and cysteine (blue) residues; methyllanthionine (Abu-S-Ala) rings are formed between didehydrobutyrine (Dhb), derived from threonine (orange) and cysteine residues.

(50). We examined the cross-resistance of *liaF* mutants to other membrane-active bacteriocins, nisin A and garvicin KS (43). As expected, both nisin A and garvicin KS showed reduced bioactivity (4- to 32-fold) toward the mutants compared to the wild-type strain.

## DISCUSSION

Bacteriocins are a promising alternative to traditional antibiotics, as they display activity against antibiotic-resistant pathogens and have many desirable properties for the control of microorganisms. They are often produced by probiotic species with GRAS

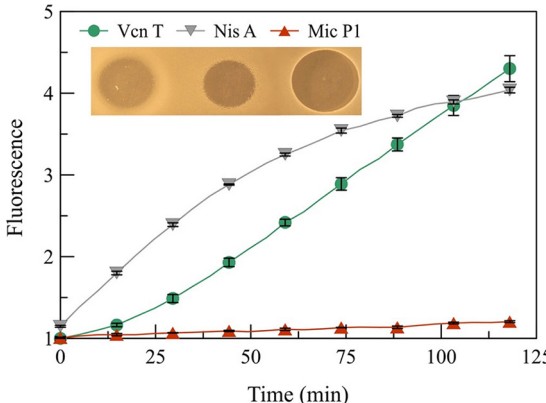

**FIG 6** Bacteriocin-induced pore formation assay. Shown is the propidium iodide fluorescence intensity over time in the presence of *E. faecium* and the antimicrobials vagococcin T (Vcn T), nisin A (Nis A), and micrococcin P1 (Mic P1). An increase in emission is observed for the pore-forming nisin A. Micrococcin P1, a non-pore-forming bacteriocin, was used as a negative control. The inhibition zone produced by each bacteriocin (2 µL) is shown at the top left.

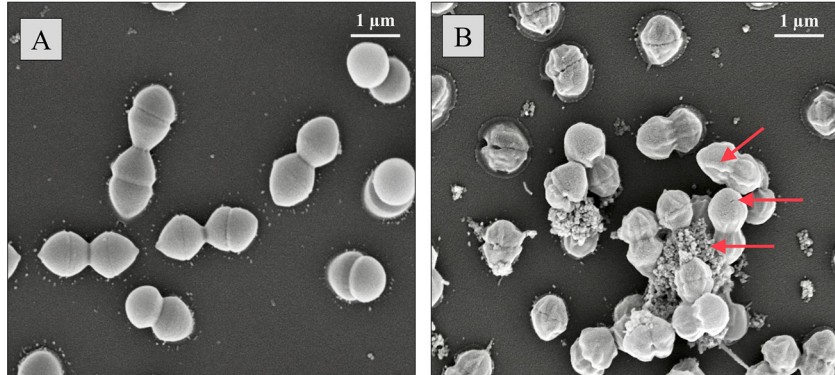

**FIG 7** Scanning electron microscopy (SEM) showing the effect of vagococcin T on *E. faecium* cells (magnification, ×30,000). Cells incubated without vagococcin T showed no visible cell damage (A), while the vagococcin T-treated cells had a shriveled appearance following a 2-h incubation with 10× MIC of vagococcin T (B). Signs of cell damage and lysis are indicated by red arrows.

(generally regarded as safe) status and have high potency and low toxicity (51). In addition, bacteriocins are arguably more amenable to biotechnological manipulation as they are defined by structural genes. Given the high potency and potential clinical applications of bacteriocins, we sought to find new bacteriocins with possible therapeutic use. To this end, we screened for bacteriocin producers in fermented fruits and vegetables that inhibited the growth of the indicator strain, a multidrug-resistant *E. faecium* isolate. From a sample of fermented maize, we successfully isolated a strain of *V. fluvialis* producing a two-peptide lantibiotic named vagococcin T (*V. fluvialis* LMGT 4216).

To our knowledge, vagococcin T is the first bacteriocin characterized from the genus *Vagococcus*. The two bacteriocin genes *vcn*A1 and *vcn*A2 are separated by a *vcn*M gene, which is an unusual arrangement: two-peptide bacteriocin genes are most often located adjacent to each other in tandem. Because of the low sequence similarity of the two vagococcin T prepeptides (20% sequence identity), each of the two *vcn*M gene products is likely dedicated to modifying its cognate bacteriocin peptide. Upstream of the bacteriocin genes in the same operon is the gene pair *vcn*FE encoding an ABC transporter that likely has a dual role in the export of the bacteriocin peptides and immunity, a property that is common for other lantibiotics, including nisin, mersacidin, and lacticin 3147 (39). At the end of the *vcn* cluster is an operon encoding proteins with homology to the Fsr quorum-sensing system from *E. faecalis*. In the Fsr system, the FsrD propeptide is exported and processed by FsrB into a small 11-amino-acid cyclic peptide pheromone. A membrane-bound sensor HPK, FsrC (VcnK), then responds to the pheromone and activates the intracellular RR FsrA (VcnR) (30). VcnQ2 and VcnQ1 show 35% and 37% sequence identities to FsrB and FsrD, respectively (Table 1). The majority of circular peptide pheromones have been reported to form a thiolactone linkage between the C-terminal amino acid (methionine, phenylalanine, or leucine) and a cysteine located 3 or 4 residues from the N-terminal cleavage site (52).

**TABLE 4** Mutations identified in *E. faecium* LMG 20705 spontaneous mutants with increased tolerance to vagococcin T

| *E. faecium* mutant | Fold increase of MIC[a] | | | Mutations[b] | Protein | RefSeq accession no. |
|---|---|---|---|---|---|---|
| | **Vcn T** | **Nis A** | **Gar KS** | | | |
| M1 | 256 | 16 | 4 | c. 323T>A; p. Ile108Asn | Stress regulator protein LiaF | WP_002328613.1 |
| M2 | 256 | 32 | 4 | c. 24dupT; p. Val9fs | Stress regulator protein LiaF | WP_002328613.1 |
| M3 | 256 | 32 | 8 | c. 422G>C; p. Trp141Ser | Stress regulator protein LiaF | WP_002328613.1 |
| M4 | 64 | 8 | 4 | c. 605T>A; p. Val202Glu | Aldose 1-epimerase | WP_002328285.1 |
| | | | | c. 514G>A; p. Gly172Arg | Metal-dependent hydrolase | WP_002287133.1 |
| | | | | c. 187A>G; p. Ile63Val | Hypothetical protein | WP_100970561.1 |
| | | | | c. 277A>T; p. Thr93Ser | Mg²⁺ cation transporter (CorA family protein) | WP_002318987.1 |

[a]Vcn T, vagococcin T; Nis A, nisin A; Gar KS, garvicin KS.
[b]c., coding DNA; p., protein; >, substitution; dup, duplication; fs, frameshift.

However, the peptide processed from FsrD contains a lactone linkage between the C-terminal methionine and the hydroxyl group of a serine residue (42). In addition, an autoinducing peptide containing a lactone ring between the C-terminal phenylalanine and a serine residue has been identified in *Staphylococcus intermedius* (53). VcnQ1 may be processed similarly, forming a lactone linkage between serine and the C-terminal phenylalanine. Interestingly, the closest homolog to VcnQ1 was found to be an unannotated ORF (159 nucleotides [nt]) in the locus of the circular bacteriocin enterocin NKR-5-3B (Ent53B) produced by *E. faecium* strain NKR-5-3 (GenBank accession number LC068607) (54). The ORF is arranged similarly to *vcn*Q1 between genes encoding an HPK and an FsrB-like protein (*orf5* and *orf6*). The predicted mature product of this ORF contains an 11-amino-acid sequence showing 73% identity (100% similarity) to the putative VcnQ1-derived pheromone. *E. faecium* NKR-5-3 produces multiple bacteriocins: enterocins NKR-5-3A, -B, -C, -D, and -Z (Ent53A, Ent53B, Ent53C, Ent53D, and Ent53Z) (55). An inducing peptide, Ent53D, has been shown to regulate the transcription of the above-mentioned bacteriocins except for NKR-5-3B (55). A derivative of the unannotated ORF in the *E. faecium* NKR-5-3 genome may be involved in the regulation of NKR-5-3B. However, it is presently not known if VcnQRK constitutes a functional quorum-sensing system in *V. fluvialis* LMGT 4216; characterization of the *vcn* regulatory system is beyond the scope of the present study.

The production of vagococcin T by *V. fluvialis* LMGT 4216 was confirmed by bacteriocin purification and MALDI-TOF MS. Vagococcin T was purified from the cell-free supernatant using a typical scheme for bacteriocin purification in our laboratory, starting with ammonium sulfate precipitation at 60% saturation (4°C), a concentration determined to be a good compromise between yield and purity for many bacteriocins. By reversed-phase chromatography, a significant increase in potency (51,200 BU/mL) was observed for the combination of fractions 23 and 28. Despite not corresponding to the two peaks in the elution profile, the noticeably higher activity observed for the combination was strong evidence of a two-peptide bacteriocin.

Mass determination of each fraction revealed single distinct peaks at 4,009.4 *m/z* and 3,181.69 *m/z* for fractions 23 and 28, respectively. Analysis of the *V. fluvialis* LMGT 4216 genome by the ribosomally synthesized and post-translationally modified peptides (RiPP) mining tool antiSMASH (30) identified a lanthipeptide gene cluster encoding two putative lanthipeptide precursors. In addition to predicting lanthipeptide genes, antiSMASH predicts the leader cleavage site, dehydrations, cross-links, and expected masses. The mass predicted for Vcn T$\alpha$ (4,010.6 Da), assuming one unmodified serine or threonine residue, corresponded well with the measured value of 4,009.4 *m/z*. However, the mass predicted for Vcn T$\beta$ (3,111.6 Da) was approximately 71 Da lower than the mass obtained by MALDI-TOF MS. The reason for this discrepancy is likely inaccurate leader peptide prediction. The predicted Vcn T$\alpha$ leader peptide is a typical double-glycine-type leader with a GG| cleavage site, while the Vcn T$\beta$ leader cleavage site was predicted to be (G)GA|. The predicted mass of Vcn T$\beta$ with the addition of alanine is 3,181.5 Da, which is consistent with the measured mass of *m/z* 3,181.67. The close correspondence between the measured and the theoretical masses provides strong evidence that the purified bacteriocin vagococcin T is the gene product of *vcn*A1 and *vcn*A2. The predicted structures of Vcn T$\alpha$ and Vcn T$\beta$ peptides are consistent with the structures of other two-peptide lantibiotics (Fig. 5).

The $\alpha$-peptide of most two-component lantibiotics employs lipid II as a docking molecule to exert its antimicrobial activity (56, 57). A lipid II-binding motif was found in Vcn T$\alpha$ (Fig. 2B), suggesting a lipid II-dependent mode of action of vagococcin T. It is believed that the $\beta$-peptide of lipid II-targeting two-component lantibiotics binds to the complex formed between lipid II and the $\alpha$-peptide, which then leads to pore formation. The predicted mode of action involving pore formation was consistent with SEM showing *E. faecium* with a shriveled appearance, lysed cells, and cell debris following exposure to vagococcin T (Fig. 7). The extracellular matrix-like material likely consists of cell debris cross-linked by the fixing agent. The pore formation property is further supported by the fact that Vcn T

showed a pore-forming ability comparable to that of nisin A, a known pore-forming lantibiotic (58, 59).

For many lantibiotics, the type of immunity system appears to correlate with the mode of action of the lantibiotic (12, 13). It is believed that producers of pore-forming lantibiotics require both the LanI and LanFE(G) components for immunity (13, 60). However, no LanI component was immediately apparent in the *vcn* cluster despite the evident pore-forming mode of action of vagococcin T (Fig. 6). Upon further analysis, a small ORF was found downstream of *vcn*T, encoding a predicted transmembrane, cationic, 50-amino-acid protein (charge 5 at pH 7). The protein sequence shows no homology to known proteins but shares similar properties with PepI, EciI, and LasJ (LanI components of Pep5, epicidin, and lactocin S, respectively), all predicted transmembrane proteins 57 to 69 amino acids long with a charge of 4 to 6 (at pH 7). Due to this similarity, we believe that this ORF is involved in lantibiotic immunity, and it is thus named *vcn*I.

Upon challenging the *E. faecium* indicator strain with the bacteriocin, we observed resistant cells at a frequency of $8.7 \times 10^{-7}$. Three randomly selected isolates with the highest tolerance to vagococcin T all had mutations in *liaF*, a negative regulator (repressor) of the LiaRS (lipid II-interacting antibiotic response regulator and sensor) cell envelope stress response system. Previous studies have shown that membrane-active antimicrobials decouple the repression by LiaF, allowing the HPK LiaS and its cognate RR LiaR to trigger genes involved in resistance (61). The effect of the genetic disruption of *liaF* is likely similar to that of the decoupling of LiaF-mediated repression. Orthologs of the Lia system exist in most *Firmicutes*, and all systems investigated so far regulate the expression of genes that protect the cell against perturbations in the cell envelope (50). In *Bacillus subtilis*, the LiaFSR system is one of the primary response systems against lipid II-interacting antibiotics such as vancomycin and bacitracin (62) but is also induced by cationic antimicrobial peptides, organic solvents, and detergents (63–65). The genes regulated by the Lia system vary between species; in *Staphylococcus aureus*, the LiaRS homolog (VraSR) upregulates genes encoding penicillin-binding proteins and proteins involved in teichoic acid synthesis, chaperones, and membrane lipid biosynthesis that together confer resistance to $\beta$-lactam antibiotics (66–69). Even though the LiaFSR regulon in enterococci remains unknown, the LiaFSR system has been implicated in resistance to daptomycin and antimicrobial peptides due to the redistribution of cardiolipin microdomains away from the division septum (70, 71). All *liaF* mutants displayed low-level cross-resistance to nisin A, another lipid II-interacting lantibiotic (Table 4). These results confirm the role of LiaFSR in mediating resistance to vagococcin T, which further supports the lipid II-mediated mode of action of the bacteriocin.

The appearance of vagococcin T-resistant colonies of *E. faecium* exemplifies the hardiness of enterococcal populations. Combination therapies will likely be needed to effectively control enterococcal populations in the future. Formulations combining bacteriocins with different modes of action have been developed and showed increased potency and a broader inhibition spectrum with a very low frequency of resistance (72, 73).

In summary, in this work, we describe the isolation and characterization of a new two-component lantibiotic, vagococcin T, showing a broad antimicrobial spectrum against Gram-positive species, including multidrug-resistant strains. Furthermore, we show that mutations in the *liaF* gene confer resistance to vagococcin T and other antimicrobials. This connection highlights LiaF and the stress response system as appealing targets for future drug development and combination therapies. Further work is required to establish the potential of vagococcin T as a therapeutic in human or veterinary medicine.

## MATERIALS AND METHODS

**Bacterial strains and growth conditions.** The indicator strain *E. faecium* LMG 20705 (FAIR-E 102) was obtained from the LMG collection (BCCM/LMG Bacteria Collection, Laboratorium voor Microbiologie, Universiteit Gent, Ghent, Belgium). *E. faecium* LMG 20705 was grown in M17 broth supplemented with 0.5%

(wt/vol) glucose (GM17) and incubated at 37°C without shaking. All other bacterial strains were grown in brain heart infusion (BHI) broth at 30°C without shaking.

**Screening for bacteriocin producers.** A selection of 40 different fruits and vegetables was purchased from a local market (Oslo, Norway) and prepared as described previously (74). Samples were screened for bacteriocin producers using a multilayer soft-agar technique. Briefly, 10-fold serial dilutions of samples were prepared in sterile saline. An aliquot (10 mL) of each dilution was mixed with 5 mL of BHI soft agar (0.7% [wt/vol] agar), plated onto a BHI agar plate (1.5% [wt/vol] agar), and allowed to solidify. A second layer of BHI soft agar was poured on top, and the plates were incubated overnight at 30°C. Next, a culture of the indicator strain grown overnight was diluted 1:100 in 5 mL BHI soft agar and poured over the plate. After an additional incubation at 30°C overnight, colonies showing a clear zone of inhibition were restreaked to obtain pure cultures. The pure culture was retested against the indicator strain before being stored in 20% glycerol at −80°C for later use.

**DNA sequencing and repetitive element PCR fingerprinting.** Genomic DNA was isolated and purified using a GenElute bacterial genomic DNA kit (Sigma-Aldrich, St. Louis, MO, USA) according to the provided protocol. The 16S rRNA gene was amplified using the universal primers 11F (5′-TAACACATGCAAGTCGAACG-3′) and 4R (5′-ACGGGCGGTGTGTRC-3′). The PCR product was purified using a NucleoSpin gel and PCR cleanup kit (Macherey-Nagel, Düren, Germany) according to the manufacturer's instructions and sent to Eurofins Genomics for Sanger sequencing. Repetitive element PCR (rep-PCR) fingerprinting was performed using primers ERIC1R (5′-ATGTAAGCTCCTGGGGATTCAC-3′), ERIC2 (5′-AAGTAAGTGACTGGGGTGAGCG-3′), and LL-rep1 (5′-TACAAACAAAACAAAAAC-3′) as previously described (74, 75).

Whole-genome sequencing was performed by BGI (Beijing Genomics Institute) (Beijing, China) using the DNBSeq sequencing platform (150-bp paired-end reads). Reads were error corrected and assembled using SPAdes v3.14.1 (76). The resulting contigs were submitted to antiSMASH and BAGEL4 for the identification of potential bacteriocin genes (29, 30). For submission, the whole-genome assembly was assembled using Unicycler v0.5.0 and annotated using the NCBI prokaryotic genome annotation pipeline (PGAP) (77, 78).

**Bacteriocin purification.** *V. fluvialis* LMGT 4216 was cultivated in 1 L of BHI broth at 30°C for 24 h. Cells were removed by centrifugation (10,000 × *g* for 30 min at 4°C), and the bacteriocin was precipitated from the culture supernatant with ammonium sulfate (60% saturation at 4°C overnight). The precipitate was harvested by centrifugation (15,000 × *g* for 40 min at 4°C), redissolved in 700 mL of distilled water, and adjusted to pH 3.5 with 1 M hydrochloric acid. The sample was applied to a Hi-Prep 16/10 SP-XL column (GE Healthcare, Chicago, IL, USA). Unbound material was washed from the column with 150 mL of 25 mM sodium citrate-phosphate buffer (pH 3.5). The bacteriocin was eluted with 100 mL of 0.5 M sodium chloride, and the eluate was then applied to a 1-mL Resource RPC column (GE Healthcare) connected to an Äkta purifier system (Amersham Pharmacia Biotech, Amersham, UK). The column was previously equilibrated with 0.1% (vol/vol) trifluoroacetic acid (TFA), and the bacteriocin was eluted from the column using a linear gradient (40 column volumes [CV]) of isopropanol containing 0.1% (vol/vol) TFA at 1 mL/min.

**Bacteriocin activity assays.** Bacteriocin activity was assayed in microtiter plates as previously described (79). A culture of the indicator strain *E. faecium* LMG 20705 or mutants grown overnight was diluted 50-fold in GM17 broth containing 2-fold dilutions of the sample to a total volume of 200 $\mu$L. The plate was incubated at 37°C for approximately 4 h, after which the absorbance at 600 nm was measured using a SPECTROstar Nano plate reader (BMG Labtech, Ortenberg, Germany). Bacteriocin activity was expressed in bacteriocin units (BU) per milliliter: 1 BU is the amount of bacteriocin that inhibits the growth of the indicator strain by at least 50% in 200 mL of culture (79). Nisin A was prepared by thoroughly resuspending milk solids containing 2.5% nisin A in 0.05% acetic acid (catalog number N5764; Sigma-Aldrich, St. Louis, MO, USA) and discarding the remaining solids by centrifugation. Micrococcin P1 was purified as previously described (73).

A spot-on-lawn assay was used to obtain the inhibition spectrum of purified vagococcin T. A vagococcin T solution was prepared by mixing fractions with the highest synergy in a 1:1 ratio. Fresh cultures grown overnight were diluted 1:100 in 5 mL of BHI soft agar and poured onto a BHI agar plate. Once the layer solidified, 2 $\mu$L of the vagococcin T solution was spotted onto the lawn. The plates were incubated overnight at 30°C, and the inhibition zones were measured.

**Propidium iodide assay.** The pore-forming mode of action of vagococcin T was investigated using a propidium iodide (PI) method (80–82). A culture of the indicator strain grown overnight was washed twice in phosphate-buffered saline (PBS) and adjusted to an optical density at 600 nm (OD$_{600}$) of 0.7 with PBS in the wells of a black microtiter plate containing 20 $\mu$M PI (final concentration) and vagococcin T. Fluorescence was measured at 5-min intervals for 2 h using a Hidex (Turku, Finland) Sense microplate reader with excitation at 535/20 nm (515 to 555 nm) and emission at 610/20 nm (590 to 630 nm). Each data point is the mean from three biological replicates, and error bars indicate ±1 SD (sample standard deviation).

**MALDI-TOF mass spectrometry.** MALDI-TOF MS was performed on an ultrafleXtreme mass spectrometer (Bruker Daltonics, Bremen, Germany) operated in reflectron mode. The instrument was externally calibrated with peptide calibration standard II (Bruker Daltonics), and positively charged ions in the range of 1,000 to 6,000 *m/z* were analyzed. The RPC-purified fractions and matrix ($\alpha$-cyano-4-hydroxycinnamic acid [HCCA]) were mixed in a 1:1 ratio and applied on a Bruker MTP 384 steel target plate (Bruker Daltonics) for analysis.

**Scanning electron microscopy.** The indicator strain was grown to mid-log phase (OD$_{600}$ of ~0.6) and incubated with vagococcin T (10× MIC) for 2 h at 37°C with gentle shaking. A culture with no bacteriocin added was used as a control. After incubation, cells were harvested by centrifugation (10,000 × *g* for 5 min),

washed twice in PBS, and resuspended in fixing solution (1.25% [wt/vol] glutaraldehyde, 2% [wt/vol] formaldehyde, PBS) for incubation overnight at 4℃. Fixed cells were then washed three times in PBS and allowed to sediment/attach on poly-L-lysine-coated glass coverslips at 4℃ for 1 h. Subsequently, attached cells were dehydrated with an increasing ethanol series (30, 50, 70, 90, and 96% [vol/vol]) for 10 min each and finally washed four times in 100% ethanol. Cells were dried by critical-point drying using a CPD 030 critical-point dryer (Bal-Tec, Los Angeles, CA, USA). Coverslips were sputter coated with palladium-gold using a Polaron Range sputter coater (Quorum Technologies, Lewes, UK). Microscopy was performed on an EVO50 EP scanning electron microscope (Zeiss, Oberkochen, Germany) at 20 kV with a probe current of 15 pA.

**Mutant analysis.** To characterize mutants of *E. faecium* LMG 20705 resistant to vagococcin T, a total of 20 plates were made as described above for the spot-on-lawn assay. However, to avoid sequencing clones of the same mutant, the lawn on each plate was prepared from genetically independent cultures (inoculated with different single colonies). Colonies that were observed at or near the center of the inhibition zone from vagococcin T following incubation overnight were picked.

Colonies from several agar plates were restreaked to obtain pure cultures. Resistance to vagococcin T was confirmed and quantified by determining the bacteriocin activity toward the mutants compared to the wild-type strain. Genomic DNA of mutant strains was isolated with a GenElute bacterial genomic DNA kit (Sigma-Aldrich, St. Louis, MO, USA) according to the manufacturer's instructions and sent to Novogene Bioinformatics Technology Co., Ltd. (Beijing, China), for sequencing (NovaSeq 150-bp paired end). Reads from the wild type were assembled using SPAdes v3.15.3 to obtain reference contigs. Snippy was used to identify variants by mapping the reads from mutant isolates to the reference contigs using default settings (83).

**Accession number(s).** The DNA sequence of the vagococcin T gene cluster was submitted to GenBank under accession number OM959625. The whole-genome shotgun project has been deposited in the DDBJ/ENA/GenBank database under accession numbers PRJNA836177 (BioProject) and SAMN28154986 (BioSample).

## SUPPLEMENTAL MATERIAL

Supplemental material is available online only.

**SUPPLEMENTAL FILE 1**, PDF file, 0.1 MB.

## ACKNOWLEDGMENTS

This work was supported by Research Council of Norway project number 275190 and by Norway Grants 2014–2021 via the National Centre for Research and Development (grant number NOR/POLNOR/PrevEco/0021/2019-00).

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
