## [Reviewer comments · Microbiology Spectrum]

Microbiology Spectrum

Identification of a Novel Two-Peptide Lantibiotic from *Vagococcus fluvialis*

Dzung Diep, Zuzana Rosenbergová, Thomas Oftedal, Kirill Ovchinnikov, Thasanth Thiyagarajah, and Martin Rebroš

Corresponding Author(s): Dzung Diep, Norwegian University of Life Sciences

Review Timeline:

Submission Date:	March 19, 2022
Editorial Decision:	May 3, 2022
Revision Received:	May 18, 2022
Accepted:	May 24, 2022

Editor: Krisztina Papp-Wallace

Reviewer(s): The reviewers have opted to remain anonymous.

Transaction Report:

DOI: <https://doi.org/10.1128/spectrum.00954-22>

May 3, 2022

Prof. Dzung B Diep
Norwegian University of Life Sciences
Laboratory of Microbial Gene Technology
Dept. of Chemistry, Biotechnology & Food Science (IKBM)
P.O. Box 5003
1432 Aas
Norway

Re: Spectrum00954-22 (Identification of a Novel Two-Peptide Lantibiotic from *Vagococcus fluvialis*)

Dear Prof. Dzung B Diep:

Link Not Available

Sincerely,

Krisztina Papp-Wallace

Journals Department
Reviewer comments:

Reviewer #1 (Comments for the Author):

The authors report on a novel two-component bacteriocin isolated from a species that was not previously reported as a bacteriocin-producer, although there was some evidence indicating the possible presence of bacteriocin-encoding genes in a recent report (doi: 10.1093/g3journal/jkaa034). The bacteriocin has a good chance to serve as a candidate for control of MDR enterococci and other health-challenging microorganisms (including Gram-negative bacteria, considering that some of the two-component bacteriocins are inhibiting them when combined with synergistically-acting antibiotics). Moreover, the species is considered by investigators as a potential probiotic for animals of agricultural importance. Therefore, the study is well-justified.

The authors used adequate methods to address the study's objective. These methods are sufficiently described in the manuscript's Materials and Methods chapter. The reviewer has just a few minor comments/suggestions which are penciled in the manuscript for the authors' convenience (see attached).

Reviewer #2 (Comments for the Author):

This is well planned, performed and written paper. Authors reports on new bacteriocins, produced by the strain belong to species that was never before reported as bacteriocinogenic. Some corrections needs to be taken into account by the authors.

Will be nice if authors can test more VRE strains in order to validate the suggested activity against different antibiotic resistance strains.

Authors have some repetitions in Results and Discussion sections. Please, try to be more focus and do not repeat the information. Moreover, in Results section try to be more focus on results and do not discuss the results. This needs to be dedicated to the discussion section.

Authors have preliminary test showing that BHI was optimal media for bacteriocin production? 60% ammonium sulphate was chosen based on preliminary ammonium sulphate % optimisation or was based on the literature? Please, consider to change the text and to be more informative.

Any reference for the method described under Ln 477-482?

Please, reference list needs additional attention. Please, pay attention to the use of italics, page numbers and volumes of the journals.

Reviewer #3 (Public repository details (Required)):

The DNA sequence of vagococcin T gene cluster is provided, but perhaps the authors should also provide the fully sequenced genome of the strains.

Reviewer #3 (Comments for the Author):

The paper submitted by Rosenbergová et al., describes the identification and characterization of a new two-peptide lantibiotic called vagococcin T. According to the authors this bacteriocin is the first one described in the genus *Vagococcus* and displays a range of activity against Gram-positive bacteria (except *S.aureus*) with notable activity against pathogens as *E. faecium*. From the point of view of the sequence, vagococcin T presents differences in sequence and structure with respect to other known two-peptide lantibiotics, especially the β -peptide. From the point of view of the action mechanism, the authors have shown a pore-forming activity and also they have characterized resistant mutants against the bacteriocin.

The paper is well written and the experiments are properly designed. However, some points should be addressed.

1) Line 90, please provide the data as supplementary information. The MIC for the antibiotics after the disk diffusion test and the rep-PCR.

2) Line 111. How did you identify the strain as *Vagococcus fluvialis*?

3) About the vagococcin gene cluster, can the author provide a figure comparing the organization of the different gene clusters of the known two-peptide lantibiotics? In figure 1, *vcnG* is indicated but is not in the gene cluster.

4) In Table 3, is it possible to provide the data as mg/L? Do you have any hypothesis about why *S. aureus* is resistant?

5) In figure 3, the tubes in which activity was observed and the peaks from the HPLC do not match. Is it ok?

6) Considering that this bacteriocin is the first described with this topology and that this work is focused on the characterization of this new bacteriocin, in figure 5 the proposed dehydration profile and ring formation pattern should be confirmed.

7) The strain full genome should be also deposited.

Staff Comments:

Preparing Revision Guidelines

- Point-by-point responses to the issues raised by the reviewers in a file named "Response to Reviewers," NOT IN YOUR COVER LETTER.

- Upload a compare copy of the manuscript (without figures) as a "Marked-Up Manuscript" file.
- Each figure must be uploaded as a separate file, and any multipanel figures must be assembled into one file.
- Manuscript: A .DOC version of the revised manuscript
- Figures: Editable, high-resolution, individual figure files are required at revision, TIFF or EPS files are preferred

Please return the manuscript within 60 days; if you cannot complete the modification within this time period, please contact me. If you do not wish to modify the manuscript and prefer to submit it to another journal, please notify me of your decision immediately so that the manuscript may be formally withdrawn from consideration by Microbiology Spectrum.

[revised manuscript text omitted]

*vcnA2* genes are genes encoding lantibiotic biosynthesis proteins *vcnM1* and *vcnM2*,
respectively. Both gene products, VcnM1 and VcnM2, showed sequence similarity with
MrsM, the modification enzyme for the lantibiotic mersacidin (36). The predicted function of
all proteins encoded in the *vcn* gene cluster is listed in Table 2.

**Figure 1** Gene organization of the vcnococcin T cluster in *V. fluvialis*. Bifunctional
 modification enzyme genes (green) are located downstream of lantibiotic precursor genes
 (blue). Lantibiotic transporter gene with leader removal function (gray) is located downstream
 of *vcnM1* and upstream from *vcnI*, encoding a potential immunity protein (yellow). Other
 genes involved in bacteriocin immunity are located at the beginning of the cluster. A group of
 genes resembling a quorum-sensing system (red) is located at the end of the cluster.

The *vcnT* gene is located downstream of *vcnM1* and encodes a C39 peptidase that
 shows 45% identity with MrsT, the mersacidin transport enzyme which cleaves the leader
 after the GG/GA motif – a typical cleavage site for many bacteriocin leaders (37). A GG-
 motif is indeed present in both VcnA1 and VcnA2 prepeptides (Fig. 2A). The mature peptides
 showed the highest homology to flavecin FlvA1a and FlvA2b peptides (42% and 46%,
 respectively) (35). Sequence alignment of Vcn T α with other lantibiotic α -peptides (Fig. 2B)
 showed that Vcn T α contain the same CTxTxEC conserved motif believed to be essential for
 lipid II docking (38). Similarly, the conserved sequence (CPTxxCt/sxxC) typical for all β -
 peptides was found in Vcn T β (Fig. 2B).

A

VcnA1

MERNPILREKKQQQLSTSGLEEVNQNIEFIENLSGG NNVVVTILQGVVGCVASWAVGNKGVCTWTVECQKNCS

VcnA2

MRTSNDIKNKTGYVEESKLEKEMIEEPDYSGG AWTTLPCIGGIIAATLNFACPTSACTKSCNK

B

α-peptide

VcnA1	NN-VWV---TILQGVVGCVASWAVGNKGVCTWTVECQKNCS-
FlvA1a	GW-KQTIIVCTIAQGTVGCLVSYGLGNNGCYCCTYTVECSKICNK
HalA1	-C--AWY-----NLSRCLGNKCACTTVECMPSCN-
LtnA1	CS-TNTF-----SISDYWGNNGAWCTTVECMAWCK-
LchA1	TITLSTC-----AIIKPLGNNGYLCTVTVECMPSCN-
PlwA1	KC--KWW-----NLSCLGNNGHVCTLSHECQVSCN-
SacA1	CS-TNTF-----SISDYWGNKGNWCTATVECMQWCK-
consensus	.. * ** * **..**..*

β-peptide

VcnA2	AW-----ITLPCIGGI-IAATLNFACPTSACTKSCNK---
FlvA2b	CSPLT-----VITITGL--I-VAAITGFDWCPTGACTYSCRV---
HalA2	CDVHAQ-----ITWPC-----ATVGVSVALCPTTKCTSQC----
LtnA2	TTPAT-----PAISI-----LSAYLSTNTCPTTKCTRAC----
LchA2	NDVNPETTPATTSSWTCI--T-AGVTVSASLCPTTKCTSRC----
PlwA2	GDPEAR-----SGIPCT--IGAAVAASIAVCPTTKCKRCKGRKK
SacA2	A-----ATG--VIGYLSNOTCPTTACTRAC----
consensus * * * * * *

**Figure 2** Predicted amino acid sequences of vagococcin T prepeptides (A). Leader sequences
 are underlined and separated from the mature peptides by a space. Multiple sequence
 alignment of α- and β-peptides of known two-peptide lantibiotics: flavecin (Flv; P0DQM1,
 P0DGL4), haloduracin (Hal), lactacin 3147 (Ltn; O87236, O87237), lichenicidin (Lch;
 P86475, P86476), plantaricin W (Plw; D2KR94, Q9AF68), and staphylococcin C55 (Sac;
 Q9S4D3, Q9S4D2) (B). The sequence alignment was performed using T-Coffee and colored
 with BoxShade; black and gray shading corresponds to identical and similar amino acids,
 respectively.

The types of immunity genes present in lantibiotic gene clusters vary, and the encoded
 immunity proteins often show little sequence identity with each other (39). Two genes of the
 LanFE(G) immunity system are present in the *vcn* cluster, *vcnF* and *vcnE*, located at the start
 of the operon. *VcnF* showed 47% identity with the ATP-binding domain NisF of the NisFEG
 transporter, and contained the conserved sequences for both Walker A and B motifs (40).

The last four genes in the cluster resembled an analog to the *Fsr* quorum-sensing system
 of *E. faecalis*; this type of quorum-sensing system has not previously been identified in other
 lantibiotic clusters (41). The product of the first ORF, designated *vcnR*, showed 39% identity
 to the response regulator (RR) *FsrA* (see Table 2). An *FsrB* homologue is encoded by the
 gene designated *vcnQ2* with 36% identity (Q for quorum). The third component, a sensor
 histidine protein kinase (HPK) encoded by *vcnK* showed 35% identity with *FsrC*. Search for
 small reading frames that could encode the pheromone component of the quorum-sensing

system revealed a small ORF between *vcnQ2* and *vcnK*. The product of this ORF gave no hits
 to any known peptides by BLAST search, however, sequence alignment showed 37% identity
 with FsrD, the gelatinase biosynthesis-activating pheromone (GBAP) prepeptide (42). It is
 therefore possible that the processed product of *vcnQ1* is a pheromone.

Another small ORF located between *vcnT* and *vcnR* also showed no sequence
 homology to known proteins by BLAST-search, however, the gene product had similar size,
 charge, and hydrophobicity as known lantibiotic immunity proteins. LanI proteins of
 comparable physicochemical properties include EciI, PepI, and LasJ, the LanI immunity
 proteins for epicidin 280, pep5, and lactocin S (39). The ORF located between *vcnT* and *vcnR*
 was therefore named *vcnI* and is further discussed in the Discussion section below.

[revised manuscript text omitted]

 has a dual role, in export of the bacteriocins peptides and immunity; a property which is
 common for other lantibiotics including nisin, mersacidin and lactacin 3147 (39). At the end
 of the *vcn* cluster is an operon encoding proteins with homology to the Fsr quorum-sensing
 system from *E. faecalis*. In the Fsr system, the FsrD propeptide is exported and processed by
 FsrB into a small 11 amino acid cyclic peptide pheromone. A membrane-bound sensor HPK
 FsrC (VcnK) then responds to the pheromone and activates the intracellular RR FsrA (VcnR)
 (30). VcnQ2 and VcnQ1 show 35% and 37% sequence identity with FsrB and FsrD,
 respectively (Table 1). The majority of circular peptide pheromones have been reported to

form a thiolactone linkage between the C-terminal amino acid (methionine, phenylalanine, or
leucine) and a cysteine located three or four residues from the N-terminal cleavage site (53).
However, the peptide processed from FsrD contains a lactone linkage between the C-terminal
methionine and the hydroxyl group of a serine residue (42). In addition, an autoinducing
peptide containing a lactone ring between the C-terminal phenylalanine and a serine residue
has been identified in *S. intermedius* (54). VcnQ1 may be processed similarly, forming a
lactone linkage between serine and the C-terminal phenylalanine. Interestingly, the closest
homologue to VcnQ1 was found to be an unannotated *orf* (159 nt) in the locus of the circular
bacteriocin enterocin NKR-5-3B (Ent53B) produced by the strain *E. faecium* NKR-5-3
(GenBank Accession: LC068607) (55). The *orf* is arranged similarly to *vcnQ1* between genes
encoding an HPK and a FsrB-like protein (*orf5* and *orf6*). The predicted mature product of
this *orf* contains an 11 amino acid sequence showing 73% identity (100% similarity) to the
putative VcnQ1-derived pheromone. *E. faecium* NKR-5-3 produces multiple bacteriocins;
enterocins NKR-5-3A, B, C, D, and Z (Ent53A, Ent53B, Ent53C, Ent53D, and Ent53Z) (56).
An inducing peptide Ent53D has been shown to regulate the transcription of the
aforementioned bacteriocins except for NKR-5-3B (56). A derivative of the unannotated *orf*
in *E. faecium* NKR-5-3 genome may be involved in the regulation of NKR-5-3B. However, it
is presently not known if VcnQRK constitutes a functional quorum-sensing system in *V.*
*fluvialis*; characterization of the *vcn* regulatory system is beyond the scope of the present
study.

The production of vagococcin T by the *V. fluvialis* isolate was confirmed by
bacteriocin purification and MALDI-TOF MS. Vagococcin T was purified from the cell-free
supernatant using a common purification scheme for bacteriocins involving ammonium
sulfate precipitation followed by cation-exchange- and reversed-phase chromatography. The
elution profile from reversed-phase chromatography showed two distinct peaks, indicating the
presence of a two-peptide bacteriocin. Indeed, when assayed individually, only fraction 28
exhibited some antimicrobial activity (400 BU/ml) against the indicator strain. However,
when all combinations of fractions were assayed (fractions 20 to 30), a significant increase in
potency (51 200 BU/ml) was observed for the combination of fractions 23 and 28. Despite not
corresponding to the two peaks in the elution profile, the high synergy observed for the
combination was strong evidence of a two-peptide bacteriocin.

Mass determination of each fraction revealed a single distinct peak at 4009.4 m/z and
3181.69 m/z for fractions 23 and 28, respectively. Analysis of *V. fluvialis* genome by the
RiPP mining tool antiSMASH (57) identified a lanthipeptide gene cluster encoding two
putative lanthipeptide precursors. In addition to predicting lanthipeptide genes, antiSMASH
predicts the leader cleavage site, dehydrations, crosslinks, and the expected masses. The mass
predicted for Vcn T α (4010.6 Da), assuming one unmodified serine or threonine,
corresponded well with the measured value of 4009.4 m/z. However, the mass predicted for
Vcn T β (3111.6 Da) was approximately 71 Da lower than the mass obtained by MALDI-TOF
MS. The reason for this discrepancy is likely caused by inaccurate leader peptide prediction.
The predicted Vcn T α leader peptide is a typical double-glycine-type leader with a GG|
cleavage site, while the Vcn T β leader cleavage site was predicted to be (G)GA|. The
predicted mass of Vcn T β with the addition of alanine is 3181.5 Da which is consistent with
the measured mass of 3181.67 m/z. The close correspondence between the measured and the

theoretical masses provides strong evidence that the purified bacteriocin vagococcin T is the
gene product of *vcnA1* and *vcnA2*. Predicted structures of Vcn T α and Vcn T β peptides are
consistent with the structures of other two-peptide lantibiotics (Fig. 5).

The α -peptide of most two-component lantibiotics employs lipid II as a docking
molecule to exert its antimicrobial activity (58, 59). A lipid II-binding motif was found in Vcn
T α (see Fig. 2B), suggesting a lipid II-dependent mode of action of vagococcin T. It is
believed that the b-peptide of lipid II-targeting two-component lantibiotics binds to the
complex formed between lipid II and the α -peptide, which then leads to pore formation. The
predicted mode of action involving pore formation was consistent with SEM showing *E.*
*faecium* with a shriveled appearance, lysed cells, and cell debris following the exposure to
vagococcin T (see Fig. 7). The extracellular matrix-like material is likely consisting of cell
debris cross-linked by the fixing agent. The pore formation property is further supported by
the fact that Vcn T showed a comparable pore-forming ability to nisin A, a known pore-
forming lantibiotic (60, 61).

For many lantibiotics, the type of immunity system appears to correlate with the mode
of action of the lantibiotic (12, 13). It is believed that producers of pore-forming lantibiotics
require both the LanI and LanFE(G) components for immunity (13, 62). However, no LanI
component was immediately apparent in the *vcn* cluster, despite the evident pore-forming
mode of action of vagococcin T (see Fig. 6). On further analysis, a small ORF was found
downstream of *vcnT*, encoding a predicted transmembrane, cationic, 50 amino acid protein
(charge 5 at pH 7). The protein sequence shows no homology to known proteins but shares
similar properties with PepI, EciI, and LasJ (LanI component of Pep5, epicidin, and lactocin
S, respectively), all predicted transmembrane proteins, 57-69 amino acids in length with a
charge of 4-6 (at pH 7). Due to this similarity, we believe this ORF to be involved in
lantibiotic immunity and is thus named *vcnI*.

Upon challenging the *E. faecium* indicator strain to the bacteriocin we observed
resistant cells with a frequency of 8.7×10^{-7} . Three randomly selected isolates with the
highest tolerance to vagococcin T all had mutations in *liaF*, a negative regulator (repressor) of
the LiaRS cell envelope stress response system (lipid II-interacting antibiotics response
regulator and sensor). Previous studies have shown that membrane-active antimicrobials
decouple the repression by LiaF, allowing the HPK LiaS and its cognate RR LiaR to trigger
genes involved in resistance (63). The effect of genetic disruption of *liaF* is likely similar to
the decoupling of LiaF-mediated repression. Orthologs of the Lia system exist in most
Firmicutes, and all systems investigated so far regulate the expression of genes that protect
the cell against perturbations in the cell envelope (51). In *Bacillus subtilis*, the LiaFSR system
is one of the primary response systems against lipid II-interacting antibiotics such as
vancomycin and bacitracin (64) but is also induced by cationic antimicrobial peptides, organic
solvents, and detergents (65–67). The genes regulated by the Lia system vary between
species; in *Staphylococcus aureus* the LiaRS homolog (VraSR) upregulates genes encoding
penicillin-binding proteins and proteins involved in teichoic acid synthesis, chaperones, and
membrane lipid biosynthesis, that together confer resistance to beta-lactam antibiotics (68–
71). Even though the LiaFSR regulon in enterococci remains unknown, the LiaFSR system
has been implicated in resistance to daptomycin and antimicrobial peptides due to the
redistribution of cardiolipin microdomains away from the division septum (72, 73). All *liaF*

mutants displayed low-level cross-resistance to nisin A, another lipid II-interacting lantibiotic
(Table 4). These results confirm the role of LiaFSR in mediating resistance to vagococcin T
which further supports the lipid II-mediated mode of action of the bacteriocin.

The appearance of vagococcin T-resistant colonies of *E. faecium* exemplifies the
hardiness of enterococcal populations. Combination therapies will likely be needed to
effectively control enterococcal populations in the future. Formulations combining
bacteriocins with different modes of action have been developed and showed increased
potency and broader inhibition spectrum with a very low frequency of resistance (74–76).

In summary, in this work, we describe the isolation and characterization of a new two-
component lantibiotic vagococcin T showing a broad antimicrobial spectrum against Gram-
positive species, including multidrug-resistant strains. Furthermore, we show that mutations
in the *liaF* gene confer resistance to vagococcin T and other antimicrobials. This connection
highlights LiaF and the stress response system as an appealing target for future drug
development and combination therapies. Further work is required to establish the potential of
vagococcin T as a therapeutic in human or veterinary medicine.
**Materials and Methods**

**Bacterial strains and growth conditions**

The indicator strain *E. faecium* LMG 20705 (FAIR-E 102) was obtained from the
LMG collection (BCCM/LMG Bacteria Collection, Laboratorium voor Microbiologie,
Universiteit Gent, Ghent Belgium). *E. faecium* LMG 20705 was grown in M17 broth
supplemented with 0.5 % w/v glucose (GM17) and incubated at 37°C without shaking. All
other bacterial strains were grown in brain heart infusion (BHI) broth at 30°C without
shaking.

**Screening for bacteriocin producers**

A selection of 40 different fruits and vegetables were purchased from a local market
(Oslo, Norway) and prepared as described previously (77). Samples were screened for
bacteriocin producers using a multi-layer soft agar technique. Briefly, 10-fold serial dilutions
of samples were prepared in sterile saline. An aliquot (10 ml) of each dilution was mixed with
5 ml of BHI soft-agar (0.7% w/v agar), plated on a BHI agar plate (1.5% w/v agar) and
allowed to solidify. A second layer of BHI soft agar was poured on top, and the plates were
incubated overnight at 30°C. Then, an overnight culture of the indicator strain was diluted
1:100 in 5 ml BHI soft agar and poured over the plate. After an additional overnight
incubation at 30°C, colonies showing a clear zone of inhibition were re-streaked to obtain
pure cultures. The pure culture was retested against the indicator strain before being stored in
20% glycerol at -80°C for later use.

**DNA sequencing and repetitive element PCR fingerprinting**

Genomic DNA was isolated and purified using a GenElute™ Bacterial Genomic DNA
kit (Sigma-Aldrich, Saint-Louis, MO, USA) according to the provided protocol. The 16S
rRNA gene was amplified using the universal primers 11F (5'-
TAACACATGCAAGTCGAACG-3') and 4R (5'-ACGGGCGGTGTGTRC-3'). The PCR
product was purified using NucleoSpin® Gel and PCR clean-up kit (Macherey-Nagel, Düren,
Germany) according to manufacturer's instructions and sent to Eurofins Genomics for Sanger
sequencing. Repetitive element PCR (rep-PCR) fingerprinting was performed using ERIC1R
(5'-ATGTAAGCTCCTGGGGATTAC-3'), ERIC2 (5'-
AAGTAAGTGACTGGGGTGAGCG-3') and LL-rep1 (5'-TACAAACAAAACAAAAC-
3') as previously described (78, 79).

Whole-genome sequencing was performed by BGI (Beijing Genomics Institute)  using
the DNBSeg sequencing platform (150 bp paired-end). Reads were error corrected and
assembled using SPAdes v3.14.1 (80). The resulting contigs were submitted to antiSMASH
and BAGEL4 for the identification of potential bacteriocin genes (29, 57).

**Bacteriocin purification**

The bacteriocin-producing strain was cultivated in 1 liter of BHI broth at 30°C for 24
448 hours. Cells were removed by centrifugation (10,000 g, 30 min, 4°C) and the bacteriocin was

449 precipitated from culture supernatant with ammonium sulfate (60% saturation, 4°C,
overnight). The precipitate was harvested by centrifugation (15,000 g, 40 min, 4°C),
redissolved in 700 ml of distilled water and adjusted to a pH of 3.5 with 1 M hydrochloric
acid. The sample was applied to a Hi-Prep 16/10 SP-XL column (GE Healthcare, Chicago, IL,
USA). Unbound material was washed from the column with 150 ml of 25 mM sodium citrate-
phosphate buffer (pH 3.5). The bacteriocin was eluted with 100 ml of 0.5 M sodium chloride,
eluate was then applied to a 1 mL Resource RPC column (GE Healthcare, Chicago, IL, USA) 456 connected to an ÄKTA purifier system (Amersham Pharmacia Biotech, Amersham, UK). The
column was previously equilibrated with 0.1% v/v TFA and the bacteriocin was eluted from
the column using a linear gradient (40 CV) of isopropanol containing 0.1% v/v TFA at 1
459 ml/min.

**Bacteriocin activity assays**

Bacteriocin activity was assayed in microtiter plates as previously described (81). An
overnight culture of the indicator *E. faecium* LMG 20705 or mutants was diluted 50-fold in
GM17 broth containing twofold dilutions of the sample to a total volume of 200 µl. The plate
was incubated at 37°C for approximately 4 hours, after which the absorbance at 600 nm was
measured using a SPECTROstar Nano plate reader (BMG Labtech, Ortenberg, Germany).
Bacteriocin activity was expressed in bacteriocin units (BU) per ml – one BU is the amount of
bacteriocin that inhibits the growth of the indicator strain by at least 50% in 200 ml of culture
(81). Nisin A was prepared by thoroughly resuspending milk solids containing 2.5% nisin A
in 0.05% acetic acid (N5764; Sigma, St. Louis, MO, USA)  and discarding remaining solids
by centrifugation. Micrococin P1 was purified as previously described (77).

Spot-on-lawn assay was used to obtain the inhibition spectrum of purified vagococcin
472 T. Vagococcin T solution was prepared by mixing fractions with the highest synergy in a 1:1
ratio. Fresh overnight cultures were diluted 1:100 in 5 ml of BHI soft-agar and poured onto a
BHI agar plate. Once the layer solidified, 2 µl of vagococcin T solution was spotted on the
lawn. The plates were incubated overnight at 30°C and the inhibition zones were measured.

**Propidium iodide assay**

The pore-forming mode of action of vagococcin T was investigated using the propidium
iodide (PI) method. An overnight culture of the indicator was washed twice in phosphate-
buffered saline (PBS) and adjusted to an OD₆₀₀ of 0.7 with PBS in the wells of a black
microtiter plate containing 20 µM PI (final concentration) and vagococcin T. Fluorescence
was measured at 5-min intervals for 2 hours using a FLUOstar OPTIMA reader (BMG
LABTECH, Ortenberg, Germany) with excitation at 535 nm and emission at 617 nm.

**MALDI-TOF mass spectrometry**

MALDI-TOF MS was performed on an ultrafleXtreme mass spectrometer (Bruker
Daltonics, Bremen, Germany) operated in reflectron mode. The instrument was externally
calibrated with peptide calibration standard II (Bruker Daltonics, Bremen, Germany) and
positively charged ions in the range of 1000 to 6000 m/z were analyzed. RPC purified

fractions and matrix [α -cyano-4-hydroxycinnamic acid (HCCA)] were mixed in 1:1 ratio and
applied on a Bruker MTP 384 steel target plate (Bruker Daltonics, Bremen, Germany) for
analysis.

**Scanning electron microscopy**

The indicator strain was grown to mid-log phase ($OD_{600} \sim 0.6$) and incubated with
vagococcin T (10x MIC) for 2 hours at 37°C with gentle shaking. A culture with no
bacteriocin added was used as a control. After incubation, cells were harvested by centrifugation
(10,000g, 5 min), washed twice in PBS and resuspended in fixing solution (1.25% w/v
glutaraldehyde, 2% w/v formaldehyde, PBS) for overnight incubation at 4°C. Fixed cells were
then washed three times in PBS and allowed to sediment/attach onto poly-L-lysine coated
glass coverslips at 4°C for 1 hour. Subsequently, attached cells were dehydrated with an
increasing ethanol series (30, 50, 70, 90, 96% v/v) for 10 min each and finally washed four
502 times in 100% ethanol. Cells were dried by critical-point drying using a CPD 030 critical
point dryer (BAL-TEC, Los Angeles, CA, USA). Coverslips were sputter coated with
palladium-gold using a Polaron Range sputter coater (Quorum Technologies, Lewes, UK).
Microscopy was performed on an EVO50 EP scanning electron microscope (Zeiss,
Oberkochen, Germany) at 20 kV and a probe current of 15 pA.

**Mutant analysis**

To characterize mutants of *E. faecium* LMG 20705 resistant to vagococcin T, a total of
20 plates were made as described for the spot-on-lawn assay. However, to avoid sequencing
clones of the same mutant, the lawn on each plate was prepared from genetically independent
cultures (inoculated with different single colonies). Colonies that were observed at or near the
center of the inhibition zone from vagococcin T following overnight incubation were picked.

Colonies from several agar plates were re-streaked to obtain pure cultures. The
resistance to vagococcin T was confirmed and quantified by determining the bacteriocin
activity towards the mutants compared to the wild type strain. Genomic DNA of mutant
strains was isolated with GenElute™ Bacterial Genomic DNA kit (Sigma-Aldrich, Saint-
Louis, MO, USA) according to manufacturer's instructions and sent to Novogene (Novogene
Bioinformatics Technology Co., Ltd, Beijing, China) for sequencing (NovaSeq 150 bp paired-
end). Reads from the wild type was assembled using SPAdes v3.15.3 to obtain reference
contigs. Snippy was used to identify variants by mapping the reads from mutant isolates to the
reference contigs using the default settings (82).

**Accession number**

The DNA sequence of vagococcin T gene cluster was submitted to GenBank under the
accession number OM959625.

**Acknowledgements**

Research council of Norway project nr 275190 and by the Norway Grants 2014-2021 via the
National Centre for Research and Development (grant number
NOR/POLNOR/PrevEco/0021/2019-00).

**References**

- 1. Werner G, Coque TM, Franz CMAP, Grohmann E, Hegstad K, Jensen L, van Schaik
531 W, Weaver K. 2013. Antibiotic resistant enterococci-tales of a drug resistance gene trafficker.
*Int J Med Microbiol* 303:360–379.
- 2. Guzman Prieto AM, van Schaik W, Rogers MRC, Coque TM, Baquero F, Corander J,
Willems RJL. 2016. Global Emergence and Dissemination of Enterococci as Nosocomial
Pathogens: Attack of the Clones? *Frontiers in Microbiology* 7:788.
- 3. Mališová L, Jakubů V, Pomorská K, Musílek M, Zemličková H. 2021. Spread of
Linezolid-Resistant Enterococcus spp. in Human Clinical Isolates in the Czech Republic. 2.
*Antibiotics* 10:219.
- 4. Weiner LM, Webb AK, Limbago B, Dudeck MA, Patel J, Kallen AJ, Edwards JR,
Sievert DM. 2016. Antimicrobial-Resistant Pathogens Associated With Healthcare-
Associated Infections: Summary of Data Reported to the National Healthcare Safety Network
at the Centers for Disease Control and Prevention, 2011–2014. *Infection Control & Hospital*
*Epidemiology* 37:1288–1301.
- 5. Cattoir V. 2022. The multifaceted lifestyle of enterococci: genetic diversity, ecology
and risks for public health. *Current Opinion in Microbiology* 65:73–80.
- 6. Noskin GA, Stosor V, Cooper I, Peterson LR. 1995. Recovery of Vancomycin-
Resistant Enterococci on Fingertips and Environmental Surfaces. *Infection Control &*
*Hospital Epidemiology* 16:577–581.
- 7. World Health Organization. 2018. Antimicrobial resistance and primary health care.
WHO/HIS/SDS/2018.56. World Health Organization.
- 8. Alvarez-Sieiro P, Montalbán-López M, Mu D, Kuipers OP. 2016. Bacteriocins of
lactic acid bacteria: extending the family. *Appl Microbiol Biotechnol* 100:2939–2951.
- 9. Zimina M, Babich O, Prosekov A, Sukhikh S, Ivanova S, Shevchenko M, Noskova S.
2020. Overview of Global Trends in Classification, Methods of Preparation and Application
of Bacteriocins. 9. *Antibiotics* 9:553.
- 10. Repka LM, Chekan JR, Nair SK, van der Donk WA. 2017. Mechanistic
Understanding of Lanthipeptide Biosynthetic Enzymes. *Chem Rev* 117:5457–5520.
- 11. Acedo JZ, Chiorean S, Vederas JC, van Belkum MJ. 2018. The expanding structural
variety among bacteriocins from Gram-positive bacteria. *FEMS Microbiology Reviews*
42:805–828.
- 12. Smits SHJ, Schmitt L, Beis K. 2020. Self-immunity to antibacterial peptides by ABC
transporters. *FEBS Letters* 594:3920–3942.
- 13. Alkhatib Z, Abts A, Mavaro A, Schmitt L, Smits SHJ. 2012. Lantibiotics: How do
producers become self-protected? *Journal of Biotechnology* 159:145–154.
- 14. Lagedroste M, Reiners J, Knospe CV, Smits SHJ, Schmitt L. 2020. A Structural View
on the Maturation of Lanthipeptides. *Frontiers in Microbiology* 11:1183.
- 15. Islam MR, Nagao J, Zendo T, Sonomoto K. 2012. Antimicrobial mechanism of
lantibiotics. *Biochemical Society Transactions* 40:1528–1533.
- 16. Piper C, Draper LA, Cotter PD, Ross RP, Hill C. 2009. A comparison of the activities
of lacticin 3147 and nisin against drug-resistant *Staphylococcus aureus* and *Enterococcus*
species. *Journal of Antimicrobial Chemotherapy* 64:546–551.
- 17. Piper C, Casey PG, Hill C, Cotter PD, Ross RP. 2012. The Lantibiotic Lacticin 3147
Prevents Systemic Spread of *Staphylococcus aureus* in a Murine Infection Model.
*International Journal of Microbiology* 2012:e806230.
- 18. Dobson A, O'Connor P m., Cotter P d., Ross R p., Hill C. 2011. Impact of the broad-
spectrum antimicrobial peptide, lacticin 3147, on *Streptococcus mutans* growing in a biofilm
and in human saliva. *Journal of Applied Microbiology* 111:1515–1523.

[revised manuscript text omitted]

75. Kranjec C, Ovchinnikov KV, Grønseth T, Ebineshan K, Srikantam A, Diep DB. 2020.
A bacteriocin-based antimicrobial formulation to effectively disrupt the cell viability of
methicillin-resistant *Staphylococcus aureus* (MRSA) biofilms. *npj Biofilms Microbiomes*
6:1–13.

76. Ovchinnikov KV, Kranjec C, Telke A, Kjos M, Thorstensen T, Scherer S, Carlsen H,
Diep DB. 2021. A Strong Synergy Between the Thiopeptide Bacteriocin Micrococcin P1 and
Rifampicin Against MRSA in a Murine Skin Infection Model. *Frontiers in Immunology*
12:2532.

77. Ovchinnikov KV, Kranjec C, Telke A, Kjos M, Thorstensen T, Scherer S, Carlsen H,
Diep DB. 2021. A Strong Synergy Between the Thiopeptide Bacteriocin Micrococcin P1 and
Rifampicin Against MRSA in a Murine Skin Infection Model. *Front Immunol* 12:676534.

78. Urbach E, Schindler C, Giovannoni SJ. 1998. A PCR fingerprinting technique to
distinguish isolates of *Lactococcus lactis*. *FEMS Microbiology Letters* 162:111–115.

79. Versalovic J, Koeuth T, Lupski JR. Distribution of repetitive DNA sequences in
eubacteria and application to fingerprinting of bacterial genomes 9.

80. Nurk S, Bankevich A, Antipov D, Gurevich AA, Korobeynikov A, Lapidus A,
Prjibelski AD, Pyshkin A, Sirotkin A, Sirotkin Y, Stepanauskas R, Clingenpeel SR, Woyke T,
Mclean JS, Lasken R, Tesler G, Alekseyev MA, Pevzner PA. 2013. Assembling Single-Cell
Genomes and Mini-Metagenomes From Chimeric MDA Products. *Journal of Computational*
*Biology* 20:714–737.

- 81. Holo H, Nilssen O, Nes IF. 1991. Lactococcin A, a new bacteriocin from *Lactococcus*
*lactis* subsp. *cremoris*: isolation and characterization of the protein and its gene. *J Bacteriol*
173:3879–3887.
- 82. Seemann T. 2015. *snippy*: fast bacterial variant calling from NGS reads.
<https://github.com/tseemann/snippy>. Retrieved 12 January 2022.

**Figure S1** Fractions (1 μ l) from reversed-phase chromatography corresponding to the first
(21 to 24) and second peak (26 to 30) were spotted individually (to the left and above black
bars) and in combination (1:1 v/v ratio) on a lawn of *E. faecium* LMG 20705. Fractions
spotted individually produced no or only small/diffuse inhibition zones, some fractions
produced large inhibition zones when spotted in combination with the largest zone produced
by a combination of fractions 23 and 28.

**Table S1** Antibiotic resistance of *Enterococcus faecium* LMG 20705.

Antibiotic	Gene ^a	Gene product	Accession number)
ampicillin ^b	-	-	-
aminoglycoside	aac(6)-Ii	aminoglycoside 6'-N-acetyltransferase	WP_002293989.1
clindamycin ^b	lnuB	lincosamide nucleotidyltransferase	WP_002294514.1
erythromycin ^b	ermB	rRNA adenine N-6-methyltransferase	WP_001038795.1
kanamycin ^b	aph(3')-IIIa	aminoglycoside O-phosphotransferase	WP_001096887.1
pleuromutilin	eatA	ABC-F type ribosomal protection protein	WP_002296175.1
spectinomycin	ant(9)-Ia	aminoglycoside nucleotidyltransferase	WP_002294509.1
streptogramin A ^{b,*}	lsaE	ABC-F type ribosomal protection protein	WP_002294513.1
streptogramin B ^{b,*}	msrC	ABC-F type ribosomal protection protein	WP_063854349.1
streptomycin ^b	ant(6)-Ia	aminoglycoside nucleotidyltransferase	WP_001255866.1
streptothricin	sat4	streptothricin N-acetyltransferase	WP_000627290.1
tetracycline ^b	tetL	tetracycline efflux MFS transporter	WP_002294500.1
	tetM	tetracycline resistance ribosomal protection protein	WP_063856394.1
vancomycin ^b /teicoplanin ^b	vanA	D-alanine-(R)-lactate ligase	WP_001079845.1
	vanHA	D-lactate dehydrogenase	WP_001059542.1
	vanRA	DNA-binding response regulator	WP_001280781.1
	vanSA	histidine kinase	WP_002305818.1
	vanXA	D-Ala-D-Ala dipeptidase	WP_000402348.1
	vanYA	D-Ala-D-Ala carboxypeptidase	WP_001812592.1
	vanZA	glycopeptide resistance protein	WP_000516404.1

790 ^a Found in *E. faecium* LMG 20705 genome with AMRFinderPlus791 ^b Tested and confirmed by disc diffusion method according to EUCAST

* Quinopristin/dalfopristin resistance

Reviewer comments:

Reviewer #1 (Comments for the Author):

The authors report on a novel two-component bacteriocin isolated from a species that was not previously reported as a bacteriocin-producer, although there was some evidence indicating the possible presence of bacteriocin-encoding genes in a recent report (doi: 10.1093/g3journal/jkaa034). The bacteriocin has a good chance to serve as a candidate for control of MDR enterococci and other health-challenging microorganisms (including Gram-negative bacteria, considering that some of the two-component bacteriocins are inhibiting them when combined with synergistically-acting antibiotics). Moreover, the species is considered by investigators as a potential probiotic for animals of agricultural importance. Therefore, the study is well-justified. The authors used adequate methods to address the study's objective. These methods are sufficiently described in the manuscript's Materials and Methods chapter. The reviewer has just a few minor comments/suggestions which are penciled in the manuscript for the authors' convenience (see attached).

Line 18: I am not very sure, but wouldn't it be more correct to say that the sequencing revealed the genes for a putative two-peptide lantibiotic?

Our response: the suggested rephrasing has been made.

Line 81: Actually, in this paper the authors also reported on the presence of the genes coding for a bacteriocin in their isolate, although the bacteriocin synthesis etc. were not confirmed.

Our response: as the reviewer points out, putative bacteriocin genes have been found bioinformatically in genomes classified within the genus *Vagococcus*, for this reason we use the term characterized as opposed to discovered or found. Characterization implies a description of its property and/or quality.

Line 172: Here and elsewhere: please, consider giving the isolate its strain identity and mention the species name along with the strain designation throughout the manuscript. This is important for the future traceability.

Our response: the isolate has been given a strain name *Vagococcus fluvialis* LMGT 4216 and submitted to our strain collection (LMGT). The full name is now used throughout the manuscript. In addition, both the sequencing reads and full assembly has been submitted to NCBI with the accession number PRJNA836177. The accession number has been included in the accession number section in the paper.

Line 241: Error bars on the graph? If error bars make the image too messy and these are the averages, please, add it in the figure legend.

Our response: this assay has now been repeated with the requested error bars included in the new figure (graph).

Line 331-336: This is purely a repetition of the results and as such does not belong to the Discussion chapter of the manuscript. Please, consider removing.

Our response: these lines have now been rephrased and shortened to be less repetitive of the results section.

Line 337: The use of the term “synergy” requires support with data on FIC or isobolograms. Perhaps, it would be more appropriate to say that the noticeably higher activity was observed in... etc.

Our response: although we disagree that using the term “synergy” requires quantitative data, the recommended change has been made.

Line 409-410: If the space allows, the authors may consider adding a few words on the possible broadening of the bacteriocin inhibition spectrum by including Gram-negatives, when used in combination with synergistically acting antimicrobials – see, for instance: doi: 10.1186/1471-2180-13-212

Our response: We are excited to explore these possibilities in future work, in the current manuscript the focus is on the discovery, isolation, and characterization of the bacteriocin.

Line 411: Data statistical analysis is missing – as applied for some assays (see comment in the manuscript).

Our response: the necessary statistical analysis/information for Figure 6 (propidium iodide assay) has been added to the materials and methods section for this assay.

Line 442: city, country

Our response: city and country has been added.

Line 447: Please, see the reviewer’s comment regarding the need for the strain designation.

Our response: the full name with strain designation is now used here.

Line 455: Please, check the journal’s requirements. For most journals, only the first time the manufacturer’s information should be delivered in full, and every next citation should contain only the name of the company.

Our response: the suggested change has

Line 469: Strictly speaking, the name of the company is Sigma-Aldrich.

Our response: the name of the company is now written in full.

Reviewer #2 (Comments for the Author):

This is well planned, performed and written paper. Authors reports on new bacteriocins, produced by the strain belong to species that was never before reported as bacteriocinogenic. Some corrections needs to be taken into account by the authors.

Will be nice if authors can test more VRE strains in order to validate the suggested activity against different antibiotic resistance strains.

Authors have some repetitions in Results and Discussion sections. Please, try to be more focus and do not repeat the information. Moreover, in Results section try to be more focus on results and do not discuss the results. This needs to be dedicated to the discussion section.

Our response: We believe it is clear from the text that no cross resistance is expected between bacteriocins like vagococcin T and antibiotics like vancomycin, due to their different target and mode of action. The focus of this work is on the discovery, isolation, and characterization of vagococcin T.

Authors have preliminary test showing that BHI was optimal media for bacteriocin production? 60% ammonium sulphate was chosen based on preliminary ammonium sulphate % optimisation or was based on the literature? Please, consider to change the text and to be more informative.

Our response: an explanation for this choice has been included in the discussion.

Any reference for the method described under Ln 477-482?

Our response: references to the publications this method has been adapted from is now added. Also changed the phrasing “the propidium iodide method” to “a propidium iodide method”, as the method used is adapted from more than one publication and not one specific described method.

Please, reference list needs additional attention. Please, pay attention to the use of italics, page numbers and volumes of the journals.

Our response: as the reviewer correctly points out, the reference list was not properly formatted. This is now fixed.

Reviewer #3 (Public repository details (Required)):

The DNA sequence of vagococcin T gene cluster is provided, but perhaps the authors should also provide the fully sequenced genome of the strains.

Our response: the full annotated genome and sequencing reads has been made public on NCBI, accession numbers are added to the manuscript.

Reviewer #3 (Comments for the Author):

The paper submitted by Rosenbergová et al., describes the identification and characterization of a new two-peptide lantibiotic called vagococcin T. According to the authors this bacteriocin is the first one described in the genus *Vagococcus* and displays a range of activity against Gram-positive bacteria (except *S. aureus*) with notable activity against pathogens as *E. faecium*. From the point of view of the sequence, vagococcin T presents differences in sequence and structure with respect to other known two-peptide lantibiotics, especially the β -peptide. From the point of view of the action mechanism, the authors have shown a pore-forming activity and also they have characterized resistant mutants against the bacteriocin.

The paper is well written and the experiments are properly designed. However, some points should be addressed.

1) Line 90, please provide the data as supplementary information. The MIC for the antibiotics after the disk diffusion test and the rep-PCR.

Our response: the antibiotic resistance status of the strain can be determined by either a MIC assay or a disc diffusion test, according to EUCAST, in our opinion doing both is redundant and adds no additional information. Also, Rep-PCR is a routine procedure which only aids in excluding clones/duplicates during screening and sequencing, the profiles provide no useful information to the reader as far as we can tell.

2) Line 111. How did you identify the strain as *Vagococcus fluvialis*?

Our response: the full annotated genome and sequencing reads has been made public on NCBI, accession numbers are added to the manuscript. NCBI Prokaryotic Genome Annotation Pipeline was used to annotate the assembly and to verify the taxonomy, the highest average nucleotide identity to type assemblies was with *Vagococcus fluvialis*.

3) About the vagococcin gene cluster, can the author provide a figure comparing the organization of the different gene clusters of the known two-peptide lantibiotics? In figure 1, vcnG is indicated but is not in the gene cluster.

Our response: A comparison of (two-peptide) lantibiotic clusters are presented in numerous publications and reviews, we feel that including clusters of many other lantibiotics would add clutter to the manuscript.

4) In Table 3, is it possible to provide the data as mg/L? Do you have any hypothesis about why *S. aureus* is resistant?

Our response: to our knowledge, there are no good and/or useful method of determining the concentration of lantibiotics without a standard. And in our experience, methods developed for proteins (Lowry Method, Bradford Assay, Qubit, NanoDrop/absorbance) are not well-suited for lantibiotics (e.g. nisin). Instead we have changed the Table text to include the volume of antimicrobial used. Regarding resistance to *S. aureus*, it is common that a bacteriocin is devoid of activity against certain species and genera, and the reason behind this is often obscure. Hence further investigation into this path requires

much extra work as well as being beyond the scope of the present study. We also feel that any hypothesis now would be pure speculation which we try to avoid.

5) In figure 3, the tubes in which activity was observed and the peaks from the HPLC do not match. Is it ok?

Our response: we mention this observation in the discussion, although the two peaks did not correspond with the bacteriocin peptides, they were present at the same stages of the elution in all purification runs. The nature behind the mismatch is not clear but we have identified the fractions containing the involved peptides, not only by bioactivity assay but by (MALDI-TOF) mass spectrometry which reveals their expected masses.

6) Considering that this bacteriocin is the first described with this topology and that this work is focused on the characterization of this new bacteriocin, in figure 5 the proposed dehydration profile and ring formation pattern should be confirmed.

Our response: we do not agree that this bacteriocin is the first described with this ring topology, the proposed ring formation is well-conserved, and the proposed structures are justifiable with the literature. In addition, the results presented in this work do not rely on the exact structure of the bacteriocin. A structure is proposed to correlate the gene products to the molecular weights determined by MALDI-TOF MS, which show a considerable agreement.

7) The strain full genome should be also deposited.

Our response: the full annotated genome and sequencing reads has been made public on NCBI, accession numbers are added to the manuscript.

May 24, 2022

Prof. Dzung B Diep
Norwegian University of Life Sciences
Laboratory of Microbial Gene Technology
Dept. of Chemistry, Biotechnology & Food Science (IKBM)
P.O. Box 5003
1432 Aas
Norway

Re: Spectrum00954-22R1 (Identification of a Novel Two-Peptide Lantibiotic from *Vagococcus fluvialis*)

Dear Prof. Dzung B Diep:

Your manuscript has been accepted, and I am forwarding it to the ASM Journals Department for publication. You will be notified when your proofs are ready to be viewed.

Sincerely,

Krisztina Papp-Wallace
Editor, Microbiology Spectrum